# Causal dependences between the coupled ocean-atmosphere dynamics over the Tropical Pacific, the North Pacific and the North Atlantic

Stéphane Vannitsem[1] and Pierre Ekelmans[2]

[1]Royal Meteorological Institute of Belgium, Avenue Circulaire, 3, 1180 Brussels, Belgium
[2]Theory of Neural Dynamics Group, Max Planck Institute for Brain Research, Max-von-Laue-Strasse 4, 60438 Frankfurt, Germany

*Correspondence to:* Stéphane Vannitsem (svn@meteo.be)

**Abstract.**

The causal dependences (in a dynamical sense) between the dynamics of three different coupled ocean-atmosphere basins, The North Atlantic, the North Pacific and the Tropical Pacific region, NINO3.4, have been explored using data from three reanalyses datasets, namely the ORA-20C, the ORAS4 and the ERA-20C. The approach is based on the Convergent Cross

Mapping (CCM) developed by Sugihara et al (2012) that allows for evaluating the dependences between variables beyond the classical teleconnection patterns based on correlations.

The use of CCM on these data mostly reveals that (i) the Tropical Pacific (NINO3.4 region) only influences the dynamics of the North Atlantic region through its annual climatological cycle; (ii) the atmosphere over the North Pacific is dynamically forcing the North Atlantic on a monthly basis; (iii) on longer time scales (interannual), the dynamics of the North Pacific and

the North Atlantic are influencing each other through the ocean dynamics, suggesting a connection through the thermohaline circulation.

These findings shed a new light on the coupling between these three different important regions of the globe. In particular they call for a deep reassessment of the way teleconnections are interpreted, and for a more rigorous way to evaluate dynamical dependences between the different components of the climate system.

*Copyright statement.* TEXT

## 1 Introduction

In environmental sciences, statistical quantities are essential tools to characterize the properties of a system, the most familiar of which are the mean, the variance, and the correlation in space or time. Correlations are very often associated with the notion of *dependences* in climate sciences, assuming that a certain variable is influencing the one to which it is correlated. Although it

is true when dealing with a Gaussian linear system, the picture becomes far more complicated when dealing with a nonlinear

system. In particular for nonlinear deterministic dynamical systems, the correlation between variables is neither sufficient nor necessary for dependence between these variables, e.g. (Granger , 1969, 2003; Sugihara et al , 2012).

Teleconnections in the form of correlations between distant points in space are used in climate sciences in order to evaluate the link of a dynamical process in some part of the world with a distant target. In particular, an important question that has attracted a lot of attention in the past decades is to know whether the Tropical Pacific system *forces* the dynamics of the climate system in the extratropics. In this context, important teleconnections are found between events like El-niño or La-niña and the temperature and precipitation patterns all over the World, e.g. (Fraedrich and Müller , 1992; Brönnimann , 2007; Lau , 2016). The origin of these teleconnections for the North Atlantic and North Pacific is currently explained through the concept of *atmospheric bridge* that allows for the transfer of information from one basin to another, e.g. (Sardeshmukh and Hoskins , 1988; Alexander et al , 2002; Yu and Lin , 2016; Lau , 2016). This explanation assumes that there is a causality principle leading to these teleconnections, mostly going from the Tropical Pacific to the remote regions. This view originating from teleconnection patterns should however be taken with care, since co-variability does not imply inter-dependences as already mentioned above. Another possible explanation of these teleconnections is the influence of an external driver on both variables that are correlated, even if they are dynamically independent.

How can we then measure this dependence? Answering this question is difficult as discussed by Clive Granger in his Nobel lecture in 2003 (Granger , 2003). One way proposed by Granger  (1969) is to use the information on the predictability of the system with or without the influence of the variable expected to be the *cause*. In this context two forecasting models should be developed one with and the other without the variable investigated as predictor (Mosedale et al., 2006; Mokhov et al , 2011; Tirabassi et al , 2015). A drawback of the approach is precisely the necessity to build such a forecasting model. Moreover as discussed in details in the supplementary material of Sugihara et al  (2012), the approach can lead to ambiguous results when applied to nonlinear deterministic dynamical systems.

A powerful method has been recently proposed by Sugihara et al (2012), known as Convergent Cross Mapping (CCM), which is a method suitable for  nonlinear deterministic dynamical systems as it is based on analogs of the current state. This method has also been tested with success when nonlinear dynamical systems are affected by noise (Mønster et al , 2017), and in coupled dynamical systems in order to identify the leading element of the coupling (BozorgMagham et al , 2015). It has also been recently used to disentangle the link between galactic cosmic rays and the variations of the global temperature (Tsonis et al , 2015; Luo et al , 2915; Ye et al , 2015) or between environmental drivers and influenza (Deyle et al , 2016). This is the method that will be used in the present study. Alternative approaches based on the transfer of information are very appealing and a lot of progress have been made in that direction (Liang and Kleeman , 2005; Runge et al , 2012; Liang , 2014, 2015). We however do not pursue in that direction and let the use of these techniques for a follow-up study.

In the present work, we address the question of causality dependence (in a dynamical sense) between the Tropical Pacific, the North Atlantic and the North Pacific coupled ocean-atmosphere dynamics at monthly to interannual time scales, in order to clarify the remote role of these different climate sub-systems on the others. This will be done by first constructing low-order systems based on projections on a few Fourier modes that are assumed to dominate the dynamics in each of the basins. This projection has already been applied successfully in the context of the analysis of the coupling between the ocean and

the atmosphere over the North Atlantic (Vannitsem and Ghil , 2017). Once these projections are identified, the time series associated with each of these modes can be analyzed using the CCM approach. The specific choice of regions and Fourier modes, although reasonable, is however a little arbitrary. The current analysis is therefore a proof-of-concept on investigating the question of dependences in the Earth System dynamics using the CCM approach, keeping in mind that an extension of the analysis to other regions and state (or phase) space representations should be explored.

Section 2 will introduce the technique of Convergent Cross Mapping (CCM). In Section 3, the datasets and the projections used are described. The datasets are coming from reanalyses performed at the European Center for Medium Range Weather Forecasts (ECMWF). The results on the application of CCM on these data are then presented in Section 4. The main conclusions and future works are outlined in Section 5.

## 2 Convergent Cross Mapping (CCM)

In the present work as in Sugihara et al (2012), two variables recorded as a function of time, say $X(t)$ and $Y(t)$, are said causally linked if they are coming from the same dynamical system. In this case a two-way *dependence* relation is present between them and the information gathered from one of the variables should, in principle, provide information on the other one. This type of dependence should not be confused with the case where a dynamical system is forced by an external driver, in which case there is a one-way dependence from the driver to the dynamical system. Disentangling the nature of this coupling is crucial in science when one is interested in describing the dynamics of the system under investigation. This is particularly true when the system is very complicated, as the Earth System is.

Several approaches have been developed in the recent past that allow for analyzing the dependences between time series. Granger Causality (GC) analysis is a celebrated approach based on the evaluation of the predictability of a variable in the absence or the presence of an hypothetical driver (Granger , 1969). As indicated in Sugihara et al (2012), its application is restricted to separable systems for which the driver can be effectively removed. In nonlinear deterministic dynamical systems in which all the variables are interconnected, the GC approach does not provide the desired answer on the effective link between the variables, see the examples given in Sugihara et al (2012). These authors therefore propose to approach the problem of causality in systems governed by deterministic dynamical systems by considering that the variables are indeed sharing the same attractor and that these variables can therefore provide information on each other. In addition, if an external driver is forcing the dynamical system under interest, the knowledge on the dynamics of the system can provide information on the driver, but not the opposite. It must be stressed here that the notion of causality is used in the very specific context of dynamical systems theory discussed above, that should not be confused with the traditional cause-effect relationship. From now on, the words causality and dependences will be used equivalently.

The original method proposed by Sugihara et al (2012) is based on the Takens' reconstruction theorem: Given a time delay $\tau$, an embedding dimension $E$ of an Euclidean space and the time series of a variable $X$, a reconstructed attractor, $M_x$, can be built. Each variable of that state space corresponds to a given delay, and each point of the reconstructed attractor is obtained from the time series as follows $\boldsymbol{X}(t) = X(t), X(t-\tau), X(t-2\tau), ..., X(t-(E-1)\tau)$. For each state space point, $\boldsymbol{X}(t)$,

of the reconstructed attractor, a set of close points are selected, called analogs, based on a distance, typically the Euclidean distance $d_i = \sqrt{\sum_j (X_j(t) - X_{j,i}(t))^2}$ where $X_j(t)$ and $X_{j,i}(t)$ are the (delay) coordinates of the reference point and the ith analog, respectively. This distance can be used provided the solutions of the dynamical system display sufficiently smooth properties (e.g. continuity) as it is often assumed for large scale geophysical fluid dynamics. Other distances could also be used. Since these are close to $\boldsymbol{X}(t)$, they should share some dynamical properties that can be exploited to make predictions starting from the current situation at time $t$. This set of analogs can also be used to recover the value of another variable, say $Y(t)$, contemporary to $\boldsymbol{X}(t)$. The idea is to use the analogs found around $\boldsymbol{X}(t)$, to *predict* the expected variable $Y(t)$, denoted $\hat{Y}(t)$, as

$$\hat{Y}(t) = \sum_{i=1}^{E+1} Y_i \times w_i \tag{1}$$

using weights defined as,

$$w_i = \frac{\exp(-\frac{d_i}{\min d_j})}{\sum_i \exp(-\frac{d_i}{\min d_j})} \tag{2}$$

where the distances $d_i$ are associated with the $E+1$ analogs obtained for the variable $\boldsymbol{X}$ (Tsonis et al , 2015), and $Y_i$ are the values of $Y$ contemporary to the ith analog on $M_x$. The number of analogs, $E+1$, is chosen such that one can form a simplex around the point $\boldsymbol{X}(t)$. The quantity $\min d_j$ denotes the minimum of $d_j$ of the $j = 1, .., E+1$ analogs around the reference point $\boldsymbol{X}(t)$. The development of this type of nonlinear forecasting traces back to several seminal papers, see for instance Casdagli (1991); Elsner and Tsonis (1992) for a detailed discussion. In their original versions, more general weights $w_i$ were proposed that should be fitted through a least square approach Casdagli (1991). Sugihara and May (1990) proposed to simplify the approach by using a simpler variant based on exponential functions depending on the distance between the analogs and the reference point. This weighting penalizes analogs that are far from the reference point, and the normalization by the minimum distance allows for having weights based only on the relative distance. This technique works well as discussed in Sugihara and May (1990). Moreover it does not need any additional parameter, implying that the approach is parcimonious.

The dynamical relation between the variables $\boldsymbol{X}$ and $Y$ can then be studied by comparing the actual value $Y(t)$ to the inferred value $\hat{Y}(t)$ obtained using analogs of $\boldsymbol{X}(t)$ on the $M_x$ attractor. Repeting this method starting from different times $t$, the correlation coefficient between $Y(t)$ and $\hat{Y}(t)$ can be computed [1]:

$$\rho = \frac{cov(Y, \hat{Y})}{\sigma_Y \sigma_{\hat{Y}}} \tag{3}$$

where $cov(.,.)$ denotes the covariance between the variables $Y(t)$ and $\hat{Y}(t)$, and $\sigma_Y$ and $\sigma_{\hat{Y}}$ are their standard deviations. The values of $\rho$ are thus lying between $-1$ and $1$, and is also known as the Pearson correlation coefficient. High values of $\rho$ indicate that the estimation of $Y$ is good. However, this correlation does not necessarily mean that there is causality, as already emphasized in the Introduction. For instance, there could be a confounding factor $Z$ that influences both $X$ and $Y$ in the same

---

[1]Note again that the correlation is measuring the linear association that could sometimes be insufficient in the case of nonlinear dependences

manner. In that case, $X$ and $Y$ would behave similarly, and therefore, there will be a correlation between $Y$ and $\hat{Y}$ (Sugihara et al , 2012).

To solve that problem, the method as described above can be repeated for increasing sizes $L$ of the samples. For each sample of size $L$, an attractor is built, from which analogs can be isolated and the correlation coefficient can be computed. Note that different ways to build these $L$-size libraries were proposed without substantial differences (Ye et al , 2015). We therefore adopt this approach proposed in Tsonis et al (2015) by randomly selecting a set of $L$ events. If $Y$ influences $\boldsymbol{X}$, the effect of $Y$ will be present in the reconstructed attractor $M_x$. By increasing the length $L$, more information on the time series of $\boldsymbol{X}$ are gathered, and therefore the selection of the analogs on the attractor is better. If there is a causality relation of $Y$ on $\boldsymbol{X}$, $\rho$ will increase with $L$.

On the contrary, if there is no causality, the added information on the variable $X$ will not give any information regarding $Y$, and the correlation coefficient will not increase with $L$. For instance, if a confounding factor $Z$ affects both $\boldsymbol{X}$ and $Y$ (that are otherwise dynamically independent of each other), they will contain a similar information, and the inference of $\hat{Y}$ based on $\boldsymbol{X}$ will display a correlation with $Y$ which is independent of $L$ (Sugihara et al , 2012). This provides a criterion on the role of a variable $Y$ on $\boldsymbol{X}$. Another important behavior of $\rho(L)$ as a function of $L$ is that the rate of increase is related with the strength of coupling. Note also that in an ideal context where the attractor can be reconstructed with precision and for $L$ going to infinity, the correlation should converge to 1. In practical situations, this precision and the asymptotic limit are never reached. The convergence is then limited to a certain level by the presence of observational error, the approximation of the dynamics (like when a low-dimensional approximation is made of the full system) and the length of the series, $L$.

The CCM method requires the knowledge of the embedding dimension $E$ and the time delay necessary for reconstructing the attractor $M_x$ from the time series. Estimating the embedding dimension based for instance on the estimates of the correlation dimension of the attractor is very challenging when the expected embedding dimension is high since the approach needs to select close analogs to work properly (e.g. Kantz and Schreiber , 1995). It therefore needs very long time series that are usually not affordable (Van den Dool , 1994; Nicolis , 1998). So a way to overcome this problem is to increase progressively the embedding dimension and see whether the results are robust or not. For the delay $\tau$, one usually uses a time period for which successive situations become sufficiently decorrelated, but not too much. Different methods are usually proposed to evaluate this delay, for instance based on decorrelation times, or simply by trial and error (e.g. Casdagli , 1991; Parker and Chua , 1989). In the present cases these delays should be relatively short for the atmosphere, but much longer for the ocean as it can be guessed by inspecting the time series of the right panels of Figures 1 and 2. For the latter we are therefore facing an important problem since the decorrelation time (or delay) is not substantially smaller than the length of the time series.

A practical alternative is to build an attractor from a set of contemporary variables that are relevant to the dynamics from an expert evaluation. In such a case a set of $E = N$ variables at the same time t are used as entries of $\boldsymbol{X}$ to represent the attractor (in fact a projection of the full attractor in a subspace of $N$ variables), and the analogs around a specific state space point $\boldsymbol{X}(t)$ can then be found in the same way as above. These analogs can be used to define the weights (2) that are in turn used to predict $Y(t)$. The influence of $Y$ on $\boldsymbol{X}$ can then be inferred by computing Eq (1). In order to see what is the impact of the modification of the approach it has been applied in the context of a well known system, a coupled ocean atmosphere model

of 36 ordinary differential equations developed in Vannitsem (2015), for which some results have been reported in Appendix B. One important result is the ability of the method to isolate dominant links between the projected attractor (the target of the analysis) and specific variables. The nature of a link can sometimes be directly related to terms present in the dynamical equations but not always due to the multivariate construction of the analogs on the projected attractor. Likewise the absence of relationship inferred from the CCM in the present framework does not imply that there could not be some dependences when other projections of the full state space are used. The conclusions reached are therefore dependent on the specific configuration used and other experimental designs are necessary to corroborate the conclusions. This is planned for a future investigation.

Overall this analysis demonstrates that the approach should be able to isolate important dependences between variables. It provides some confidence in the CCM algorithm, but we should keep in mind that the system explored in Appendix B is relatively simple and the application of CCM on more sophisticated climate models is worth performing. This is left for a future study whose results will be compared with the ones of the present analysis. Note also that when the series are much shorter than discussed in Appendix B, correlation can also be negative indicating that the total length of the series has an important impact. This will also be further discussed in Section 4.1.

The CCM method with this modification will be applied on the data presented in the next Section. Note that in order to evaluate the impact of the random sampling of $L$ events in the datasets, we can repeat the sampling a certain number of times and infer a mean (or a median when strong asymmetries are present) and a standard deviation (here using the Fisher Z test). In the experiments that will be described below, this approach is adopted and each correlation value is estimated over a large number of samples. The algorithm is sketched in Appendix A.

## 3  Time series based on reanalysis datasets

The dynamics of the coupled ocean-atmosphere system has been recently investigated by adopting a novel approach which finds its roots in the low-order modelling of such dynamical systems (Vannitsem , 2015; Vannitsem et al , 2015). It consists at projecting key fields of the large-scale dynamics of the system on a few sets of modes that are dominating its dynamics. In Vannitsem and Ghil (2017), the coupling between the ocean and the atmosphere over the Atlantic has been investigated by projecting the geopotential at 500 hPa on the mode $F_1 = \sqrt{2}\cos(\pi y/L_y)$, and the ocean potential temperature field at a certain depth (close to the surface) and the sea surface height on the mode $\phi_2 = 2\sin(\pi x/L_x)\sin(2\pi y/L_y)$. Note the sea surface height is a proxy for the upper-layer ocean streamfunction field. $F_1$ is one of the largest-scale Fourier modes of the atmospheric field that is confined in an $x$-periodic $\beta$-channel with free-slip boundary conditions in the $y$-direction, while $\Phi_2$ is one of the dominant Fourier modes compatible with free-slip boundary conditions in a rectangular, $L_x \times L_y$ closed basin. The latter mode corresponds to the typical structure of a double gyre in such a closed ocean basin. We thus expect the projection of the geopotential on $F_1(x,y)$ to provide information on the intensity of the large-scale eastward zonal transport in the atmosphere, while the projection of the temperature and streamfunction field in the ocean on $\phi_2(x,y)$ will allow us to evaluate the strength of the dominant component of the meridional gradient of temperature and the intensity of the double-gyre dynamics in the ocean, respectively. The domain chosen in terms of the spherical coordinates is 55°W $\leq \lambda \leq$ 15°W, 25°N

$\leq \phi \leq 60°$N, with $(x = \lambda - \lambda_0, y = \phi - \phi_0)$; here $(\lambda_0 = 305°, \phi_0 = 25°)$ and $(L_x = 40°, L_y = 35°)$. Note that the domain used here is the same as in Vannitsem and Ghil (2017) but a typographical error on the domain of projection is reported in the supplementary material of Vannitsem and Ghil (2017). The time series obtained for the North Atlantic will be denoted $NA_{\Psi_a,1}, NA_{\theta_o,2}, NA_{\eta_o,2}$ for the projection the geopotential at 500 hPa on the mode $F_1$, the ocean potential temperature field at

5 meter deep and the sea surface height on the mode $\phi_2$, respectively.

A similar approach can be performed for the North Pacific, except that the domain is now larger in the zonal direction. In this case the spherical-rectangle domain is (165°E–225°E, 25°N–60°N). The series obtained for this domain will be denoted $NP_{\Psi_a,1}, NP_{\theta_o,2}, NP_{\eta_o,2}$ as for the North Atlantic. Note that for both basins the projected time series contains the dominant part of the variability. It however does not preclude that other important processes are missing in the description here. Further

analysis with more modes are certainly worth doing in the future.

For the Tropical Pacific one can wonder what kind of variables should be considered. First a dominant variable in the Tropical Pacific is the mean temperature of the upper ocean layer, known to be associated with the dynamics of El-Niño. It is also known that the Walker circulation is considerably affected by the upper layer ocean temperature, and vice versa (Philander , 1990). Let us therefore consider for now the mean ocean potential temperature in the NINO3.4 region, known to have strong correlation

with the variability in the North Atlantic region (Brönnimann , 2007). For the characterization of the Walker circulation, the zonal wind at 500 hPa and 200 hPa over the same domain are chosen. They provide some information on the position and the strenght of the Walker circulation over the NINO3.4 region. The series obtained will be denoted as $NI_{U200}, NI_{U500}, NI_{\theta_o,av}$.

This approach of reducing the dynamics of the ocean and the atmosphere to a few spectral large-scale components may at first sight look arbitrary. However for the two midlatitudes basins these modes possess the largest amplitudes (Vannitsem

and Ghil , 2017), and for the Tropical Pacific, it is known that these large scale flows are strongly affected by the interaction between the ocean and the atmosphere. Moreover we are interested in the basin scale interaction between midlatitudes and the tropics. If such an interaction exists, we expect that these should be visible through the analysis of these large-scale fields. It is clear that these specific variables do not represent the full dynamics, and additional analyses with more modes is worthwhile, in particular to see what is the role of the main currents present in the ocean like the Gulf Stream or the Kuroshio current.

Three different reanalyses datasets from the European Center for Medium-Range Weather Forecasting (ECMWF) are used. The ERA-20C dataset provides a continuous reanalysis for the atmosphere of the 20th century which assimilates observations from surface pressure and surface marine winds only. It is produced using the IFS model cycle Cy38r1 and detailed information can be found on the website of the ECMWF at https://www.ecmwf.int/en/forecasts/datasets/reanalysis-datasets/era-20c, see also the report on the quality of this reanalysis dataset (Poli et al , 2015). It covers the period 1900-2010.

The second dataset is the Ocean reanalysis ORAS4 obtained using the NEMO model. The ocean model is forced by the heat, momentum and fresh water fluxes at the upper surface, and ocean observations. For the upper surface fluxes, the ERA40 reanalysis dataset is used from September 1957 to December 1988, then the ERA-Interim from January 1989 to December 2009. In 2010, these fluxes are provided by the ECMWF operational analyses. For the SST and ice products, ERA40 (until December 1981) and Reynolds dataset are used. Finally observational data within the ocean are the temperature and salinity

profiles from September 1957 to December 2010, and the sea level anomalies from November 1992 onward. More information

on the datasets used and the model configuration can be found at tps://www.ecmwf.int/en/research/climate-reanalysis/ocean-reanalysis, and more information on the quality of this product can be found in Balmaseda et al. (2013). The period covered by the dataset used in the present study is fixed from January 1958 to December 2010.

Finally, the third reanalysis dataset used is the ORA-20C which is a 10-member ensemble of ocean reanalysis covering
the 20th century using atmospheric forcing from ERA-20C. This dataset is more homogeneous than the ORAS4 since the atmospheric forcing is consistent during the whole 20th century. As pointed out in de Boisséson and Balmaseda (2016), the uncertainty is large during the first part of the century before the assimilation process constrains all the members of the ensemble to a state more consistent with other reanalysis products. One can suspect this dataset to be better than the ORAS4 for the dynamics of the ocean during the second part of the century since the state of the ocean has gotten some time to adjust
toward a representative climatology during the first half. We will therefore use data from January 1958 consistently with the ORAS4 dataset to December 2009, the final date of data availability at the time when the present work has been conducted.

So the atmospheric data from the ERA-20C that will be used in the present work will cover the same periods as the ones fixed for the ORAS4 and the ORA-20C, respectively. All data used are monthly values.

The different monthly-averaged time series obtained by projecting the fields on the 2 Fourier modes are grouped by zones,
3 for the North Atlantic (containing one series for the atmosphere and two series for the ocean), three for the North Pacific (as for the North Atlantic), and three for the Tropical Pacific (two series for the atmosphere and one series for the ocean). The nine time series based on the reanalyses ERA-20C and ORA-20C are displayed in Fig. 1 for the three regions. The three series in each zone will constitute a 3-dimensional projection of the local coupled ocean-atmosphere dynamics. The same projections but using the ERA-20C and the ORAS4 are displayed in Fig. 2.

Let us first briefly investigate the covariance structure of these time series. Table 1 displays the covariances between the different time series of Figs. 1 and 2, with on the left side of each column the covariances for the series of Fig. 1, while on the right side, the ones corresponding to series of Fig. 2. There are a few remarkable correlations. First the ones between the atmospheric fields $NA_{\Psi_{a},1}$, $NI_{U200}$, $NI_{U500}$ and $NP_{\Psi_{a},1}$, suggesting that some key variables of the global dynamics have been selected. The ocean temperature modes, $NA_{\theta_{o},2}$, $NP_{\theta_{o},2}$ and $NI_{\theta o,av}$ are also highly correlated. Interestingly the $NP_{\Psi_{o},2}$
is anti-correlated with $NI_{\theta o,av}$. Another remarkable result is that the transport in the North Pacific, $NP_{\Psi_{o},2}$, is highly correlated with the transport in the North Atlantic, $NA_{\Psi_{o},2}$, although different amplitudes are found for the two ocean reanalysis datasets. Some other correlations are much less robust when one looks at the two different ocean datasets, in particular associated with the transport and the ocean temperature over the North Atlantic (second and third column in Table 1). These differences should be associated with the different approaches to force the ocean model, and reflect important uncertainties in reconstructing the
past evolution of the Earth System.

Important correlations appear in the datasets explored, suggesting that common information are present in the different coupled ocean-atmosphere basins discussed here. These correlations are presumably highly dependent on the seasonal cycle affecting the Earth system. Further analysis by removing the seasonal signal could be done to clarify the correlations between the anomalies found in each basins. We will not, however, go in that direction in the present work since there are several ways
to do it and since the seasonal signal is part of the dynamics itself. It suffices here to recognize that links exist between these

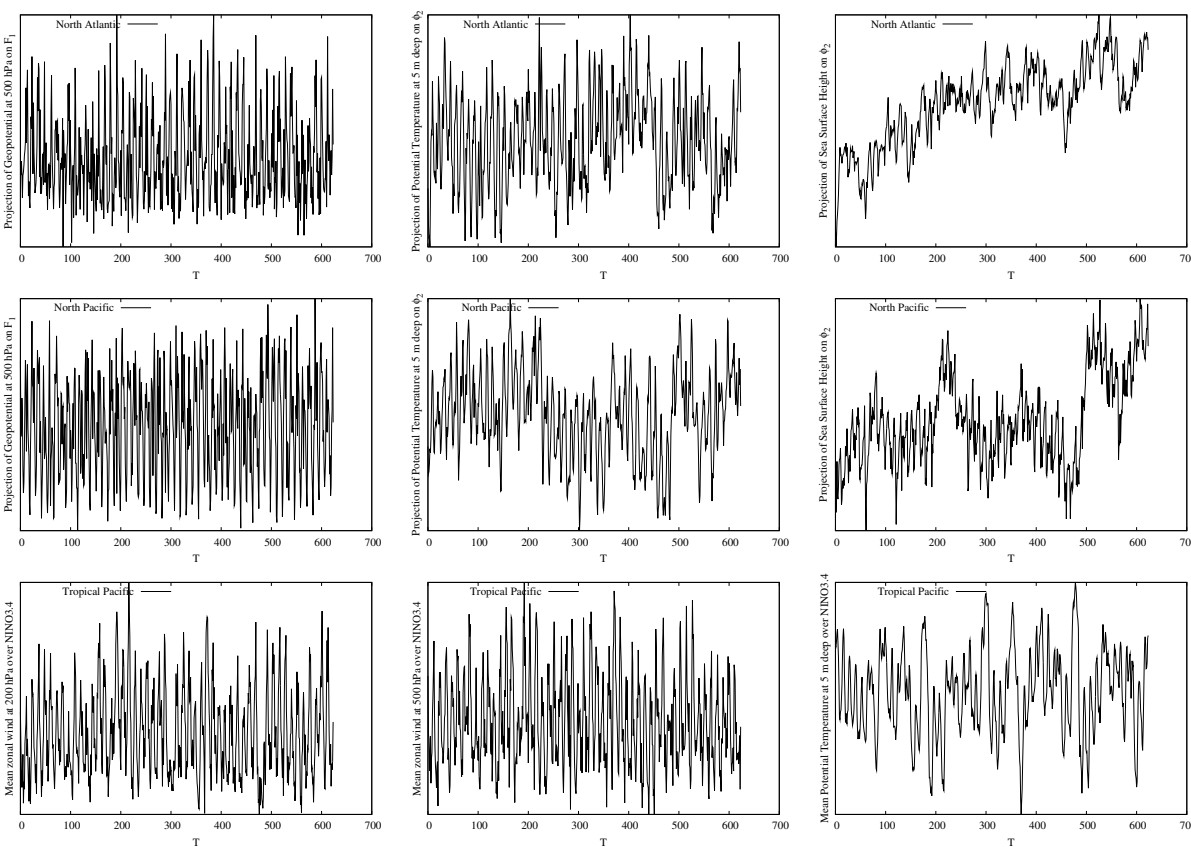

**Figure 1.** Monthly-averaged time series of the projections of the Atlantic, Pacific and Tropical fields on the dominant modes of the dynamics, as obtained from the ocean reanalysis ORA-20C and the ERA-20C atmosphere reanalysis. Top row from left to right, the geopotential at 500 hPa projected on $F_1$, the ocean potential temperature at 5 meters deep projected on $\phi_2$, and the sea surface height projected on $\phi_2$ for the Atlantic. Middle row, as for the top row but for the Pacific. Bottom row from left to right, zonal velocity at 200 hPa and 500 hPa, and the ocean potential temperature at 5 meters deep averaged over the NINO3.4 region. All time series are standardized (with a mean equal to 0 and a variance equal to 1).

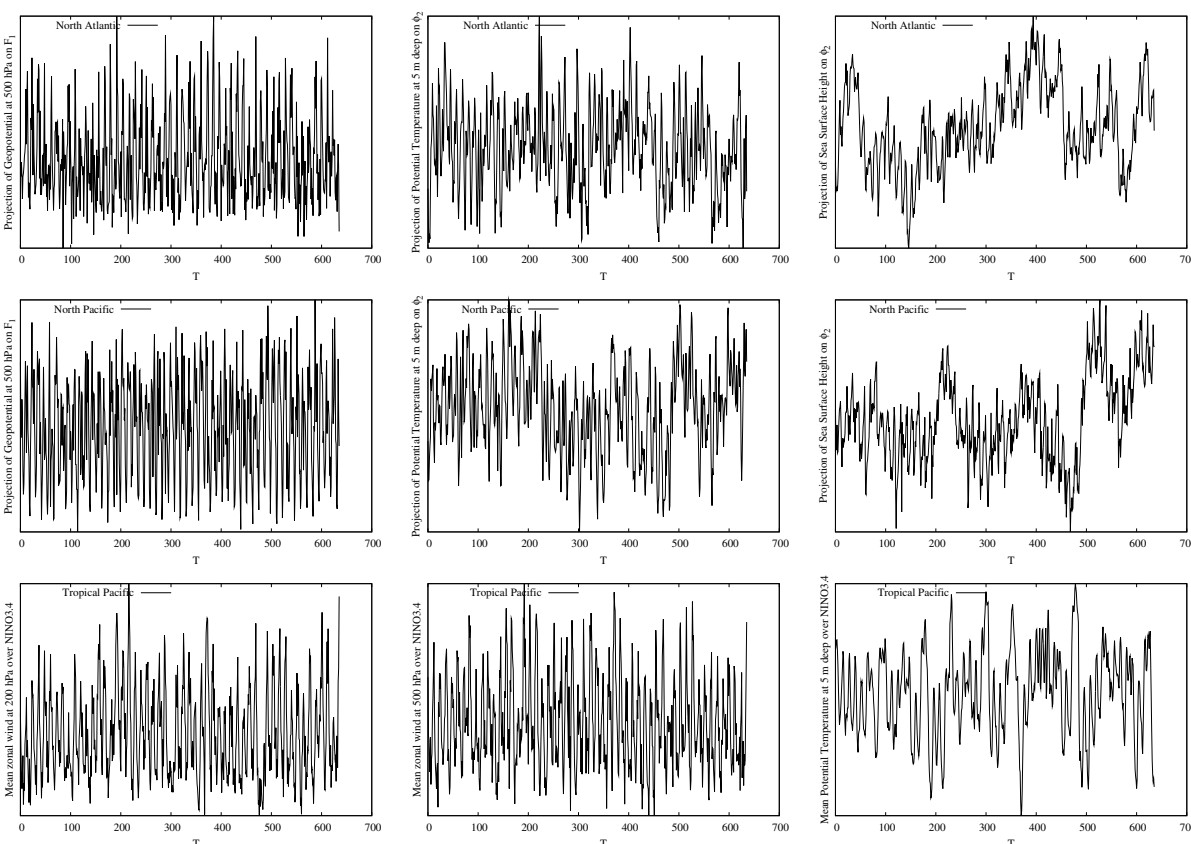

**Figure 2.** Monthly time series of the projections of the Atlantic, Pacific and Tropical fields on the dominant modes of the dynamics, as obtained from the ocean reanalysis ORAS4 and the ERA-20C atmosphere reanalysis. Top row from left to right, the geopotential at 500 hPa projected on $F_1$, the ocean potential temperature at 5 meters deep projected on $\phi_2$, and the sea surface height projected on $\phi_2$ for the Atlantic. Middle row, as for the top row but for the Pacific. Bottom row from left to right, zonal velocity at 200 hPa and 500 hPa, and the ocean potential temperature at 5 meters deep averaged over the NINO3.4 region. All time series are standardized.

basins, whose nature will be clarified by using the CCM algorithm discussed in Section 2. It will be shown that the annual cycle is also affecting the CCM results and two different ways to disentangling its role on the causality analysis will be proposed.

## 4 Results of the application of Convergent Cross Mapping

### 4.1 Reanalyses: ERA-20C/ORA-20C

Let us start the analysis by investigating the CCM for the series displayed in Fig. 1. In Panel (a) of Fig. 3, the correlation $\rho(L)$ of the inferred variable $\hat{Y}$ with the actual solution $Y$ is shown for the three variables of the Tropical Pacific. These inferred variables are obtained by building analogs in the North Atlantic as described in Section 2. Note that a 95% confidence

**Table 1.** Correlation coefficients between the different variables for the two reanalysis datasets. On the left: ORAS4, ERA-20C; and on the right: ORA-20C, ERA-20.

| | $NA_{\Psi_{a,1}}$ | $NA_{\theta_{o,2}}$ | $NA_{\eta_{o,2}}$ | $NI_{U200}$ | $NI_{U500}$ | $NI_{\theta_{o,av}}$ | $NP_{\Psi_{a,1}}$ | $NP_{\theta_{o,2}}$ | $NP_{\eta_{o,2}}$ |
|---|---|---|---|---|---|---|---|---|---|
| $NA_{\Psi_{a,1}}$ | 1 | 0.22 ǀ 0.12 | 0.12 ǀ 0.05 | 0.45 ǀ 0.45 | 0.45 ǀ 0.46 | −0.10 ǀ −0.15 | 0.41 ǀ 0.41 | −0.18 ǀ −0.14 | 0.02 ǀ −0.03 |
| $NA_{\theta_{o,2}}$ | 0.22 ǀ 0.12 | 1 | 0.52 ǀ 0.41 | −0.06 ǀ −0.20 | −0.03 ǀ −0.17 | −0.13 ǀ −0.10 | 0.07 ǀ −0.05 | 0.39 ǀ 0.42 | 0.28 ǀ 0.29 |
| $NA_{\eta_{o,2}}$ | 0.12 ǀ 0.05 | 0.52 ǀ 0.41 | 1 | −0.17 ǀ −0.06 | −0.19 ǀ −0.13 | 0.07 ǀ 0.06 | −0.09 ǀ −0.01 | 0.07 ǀ 0.10 | 0.25 ǀ 0.55 |
| $NI_{U200}$ | 0.45 ǀ 0.45 | −0.06 ǀ −0.20 | −0.17 ǀ −0.06 | 1 | 0.82 ǀ 0.83 | −0.48 ǀ −0.58 | 0.49 ǀ 0.50 | −0.04 ǀ −0.04 | 0.07 ǀ 0.07 |
| $NI_{U500}$ | 0.45 ǀ 0.46 | −0.03 ǀ −0.17 | −0.19 ǀ −0.12 | 0.82 ǀ 0.83 | 1 | −0.40 ǀ −0.47 | 0.50 ǀ 0.51 | −0.08 ǀ −0.07 | −0.08 ǀ −0.04 |
| $NI_{\theta_{o,av}}$ | −0.10 ǀ −0.15 | −0.13 ǀ −0.10 | 0.07 ǀ 0.06 | −0.48 ǀ −0.58 | −0.40 ǀ −0.47 | 1 | −0.05 ǀ −0.12 | −0.38 ǀ −0.40 | −0.22 ǀ −0.21 |
| $NP_{\Psi_{a,1}}$ | 0.41 ǀ 0.41 | 0.07 ǀ −0.5 | −0.09 ǀ −0.01 | 0.49 ǀ 0.50 | 0.50 ǀ 0.51 | −0.05 ǀ −0.12 | 1 | −0.01 ǀ −0.01 | 0.25 ǀ 0.18 |
| $NP_{\theta_{o,2}}$ | −0.18 ǀ −0.14 | 0.39 ǀ 0.43 | 0.07 ǀ 0.10 | −0.04 ǀ −0.04 | −0.08 ǀ −0.07 | −0.38 ǀ −0.40 | −0.01 ǀ −0.01 | 1 | 0.57 ǀ 0.58 |
| $NP_{\eta_{o,2}}$ | 0.02 ǀ −0.03 | 0.28 ǀ 0.29 | 0.25 ǀ 0.55 | 0.07 ǀ 0.07 | −0.08 ǀ −0.04 | −0.22 ǀ −0.21 | 0.25 ǀ 0.18 | 0.57 ǀ 0.58 | 1 |

interval is provided based on the Fisher Z-transform test based on resampling a certain number of times the samples of length $L$, indicating that the influence of the two atmospheric Tropical series is significant. This confidence interval for large $L$ is larger than for smaller values due to the fact that we are reaching the limit of the number of data points. Note also that for this analysis, the analogs have been selected to be at least separated by a period of 12 months. Longer exclusion periods have been used without substantial differences.

An increase of the correlation is found as a function of $L$, suggesting that the North Atlantic depends on the two large-scale atmospheric variables selected for the Tropical Pacific. The correlation for the ocean temperature of the Tropical region is however negative. As already mentioned previously this feature may occur when the time series is too short and when there is no significant coupling between the variables. This suggests that the prediction of the Tropical temperature based on analogs over the Atlantic are very poor, and therefore indicates the absence of influence of the Tropical ocean temperature. So the dominant influence is from the zonal flows in the atmosphere, associated with the dynamics of the Walker circulation.

The influence of the three variables of the North Pacific on the North Atlantic is very important as shown in Panel (b) with the three CCM increasing and significantly positive. The North Pacific ocean dynamics has a larger influence than the North Pacific ocean temperature on the Atlantic, as reflected by the larger amplitude of the CCM values. Note that the reduction of the state space coordinates associated with $X$ from three to two also provides interesting results with a dominating influence from the atmospheric Tropical Pacific variables on the two-dimensional projection $(NA_{\Psi_{a,1}}, NA_{\theta_{o,2}})$. However the increase before saturation of $\rho(L)$ is much more limited than when using the three variables to build the North Atlantic projection of the attractor (not shown). The latter analysis suggests that the dependences between these regions is better elucidated based on the three dimensional space.

The impacts of the North Atlantic and of the North Pacific regions on the Tropical Pacific are displayed in Panels (c) and (d). All CCM values of Panel (c) are almost flat as a function of L suggesting that even when it is positive (very significant for the geopotential at 500 hPa), there is no dependence of the Tropical Pacific on the dynamics over the Atlantic. A slightly different picture emerges for the influence of the North Pacific on the Tropical Pacific, with a slightly increasing CCM for the Geopotential at 500 hPa over the North Pacific suggesting an influence of the upper-air dynamics over the North Pacific on the Tropical Pacific. Note that sometimes it is difficult to have a clearcut answer on the increase or not of a correlation as a function

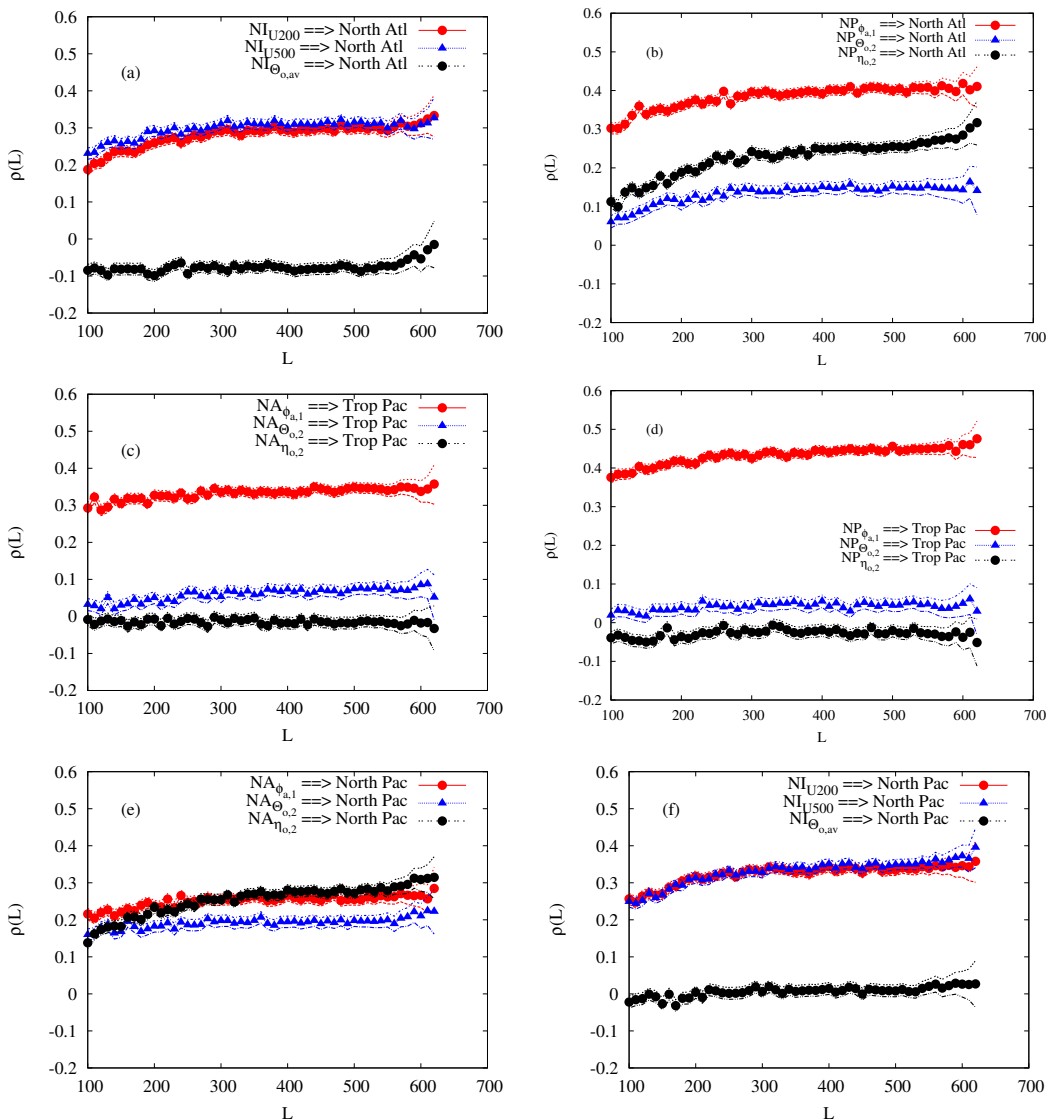

**Figure 3.** CCM as a function of the length L of the samples, as obtained from the monthly time series displayed in Fig. 1 for the reanalyses ERA-20C and ORA-20C. Each line with symbols corresponds to the influence of one variable on a specific coupled ocean-atmosphere basin. The specific variables are denoted in the caption corresponding to each line in each Panel.

of $L$. There is a degree of arbitrariness that should be alleviated using other approaches as the ones that will be discussed later based on the temporal averaging or based on surrogates.

In Panels (e) and (f), the CCM values characterizing the influence of the North Atlantic (e) and Tropical Pacific (f) on the North Pacific, are displayed. A clear increase of CCM associated with the influence of the North Atlantic ocean dynamics on the North Pacific is found (Panel (e)). The CCM values for the two other variables are also slightly increasing with $L$, suggesting a dependence of these two variables on the North Pacific. Interestingly the influence of the Geopotential of the North Pacific on the North Atlantic (Panel (b)) is more important than the one from the North Atlantic to the North Pacific since the correlation is higher. This asymmetry looks reasonable since the dominant flow is eastward in the Northern extratropics. Finally Panel (f) suggests that the atmospheric zonal flow over the Tropical Pacific influences the North Pacific dynamics, but not the ocean temperature.

To summarize, the analysis reveals that: (i) The upper-air Tropical Pacific dynamics and the North Pacific ocean and atmosphere dynamics influence the dynamics over the North Atlantic; (ii) the upper-air North Pacific dynamics influences the Tropical Pacific; and (iii) the North Atlantic ocean dynamics and the upper-air Tropical Pacific dynamics influence the dynamics over the North Pacific. These results seems to support the view of several authors, e.g. Sardeshmukh and Hoskins (1988); Alexander et al (2002); Lau (2016), that there is an *atmospheric bridge* dependence from the Tropical Pacific to the North Atlantic, via the North Pacific. Furthermore, a dependence between the North Pacific and the North Atlantic emerges, which is mostly oriented from the North Pacific to the North Atlantic since the correlations are larger in Panel (b) than in Panel (e) for the atmospheric variable, in line with the findings of Drouard et al (2015). But another very interesting dynamical signature emerges suggesting that the North Pacific and the North Atlantic ocean dynamics are dependent to each other. A possible candidate for this coupling is the thermohaline planetary circulation that affects both regions of the globe.

To disentangle the importance of the thermohaline circulation which displays a variability on very long time scales, one can investigate the dependence properties when longer time scales are considered. An average of the data set has been performed using a sliding window of 12 months. This approach allows for removing most of the impact of the annual signal, while keeping a number of data points large enough to perform the CCM analysis. Larger windows could be used but it will introduce very long time correlations that could penalize the selection of analogs since one needs analogs that are sufficiently uncorrelated in time. The results are displayed in Fig. 4. A first remarkable result is the fact that for a one-year average, the mutual dependence of the dynamics over the North Atlantic and Pacific is dominated by the ocean dynamics (see Panels (b) and (e)). The CCM values of the other variables in these two Panels are flat, and close or below 0. At the same time new dependences emerge between the North Pacific and the Tropical Pacific (Panel (d)), in particular for the ocean transport. This further supports the conjecture that the three regions are coupled via the large scale global ocean dynamics. This is presumably linked with the thermohaline circulation, but further analyses using additional variables and Tropical regions are needed in order to disentangle this point. This will be the subject of a future work.

Another very important finding in Fig. 4 is the fact that CCM values are now close to 0 (and do not display any dependence in L) for the influence of the Tropical Pacific on the North Atlantic, and vice versa (Panels (a) and (c)). It suggests that the dependence between these two regions is confined to time scales smaller than a year. One can therefore wonder whether this

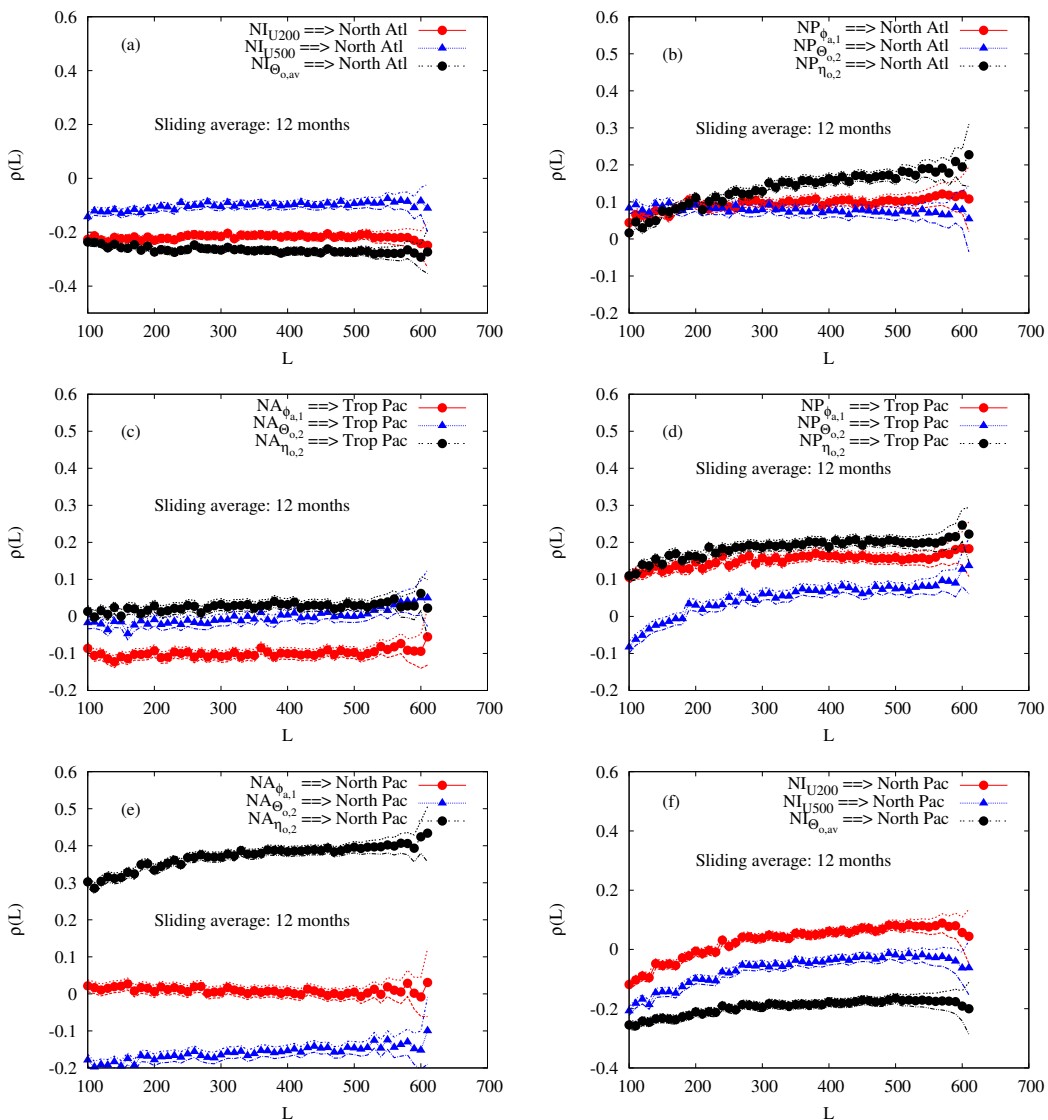

**Figure 4.** CCM as a function of the length L of the samples, as obtained from the series of Fig. 1 after averaging over a sliding window of 12 months for the reanalyses ERA-20C and ORA-20C. Each line with symbols corresponds to the influence of one variable on a specific coupled ocean-atmosphere basin. The specific variables are denoted in the caption corresponding to each line at each Panel.

dependence is associated purely to the annual cycle or to some specific Tropical events like El-Niño or La-Niña, or in other words if it is mostly associated with the climatology of the Tropical Pacific or not.

To clarify this point, surrogate 3-dimensional attractors have been built by using the monthly means (averaged over the different years) on which random anomalies with the appropriate variance are superimposed. These random anomalies were simulated assuming a gaussian distribution around each monthly mean. The variance of the distribution is estimated using the anomalies of the corresponding month for all years in the datasets.

These new attractors are then used to predict the true variable of interest $Y$. Figure 5 displays the corresponding Panels that should be compared to Fig. 3. Two different random surrogates have been built, implying that two curves are displayed in each Panel for each variable, $Y$. The first remarkable result is that the CCM values of the ocean dynamics variables of Panels (b) and (e) are now flat and close to 0, suggesting that the CCM values for the transport in the two North ocean basins found in Fig. 3 are indeed indicating a dynamical coupling between the two basins beyond the annual climatological variations. Also in Panel (b), the CCM values for the influence of the North Pacific series (ocean temperature and Geopotential at 500 hPa) on the surrogate attractor of the Atlantic considerably decrease, suggesting the importance of the influence of the North Pacific on the North Atlantic beyond the annual climatological influence.

However when looking at the results in Panel (a), the CCM values based on the use of the surrogate attractors are very close to the one obtained with the actual attractors. This surprising feature suggests that there is no influence between the Tropical Pacific and the North Atlantic beyond the annual climatological variations. Or in other words that the Tropical Pacific variability does not influence the anomalies over the North Pacific. This has a very strong implication in the sense that there is no dynamical link between an event like El Niño or La Niña and the anomalies over the North Atlantic. A similar picture is found for the influence of the Tropical Pacific on the North Pacific as illustrated in Panel (f).

Finally to further test the robustness of these results a complementary way to clarify whether monthly anomalies between different basins are indeed related to each other is to apply direcly CCM on these anomalies. The results are displayed in the Appendix C, Fig. C1 The same conclusions are reached with the absence of dependences between the variables over the North Atlantic basin and the Tropical Pacific (panels (a) and (c)), and a strong mutual dependence between the ocean dynamics over the North Atlantic and the North Pacific (panels (b) and (e)).

In summary, in the limit of the data at our disposal, the analysis suggests that the anomalies associated with the dynamics over the North Atlantic and North Pacific cannot be inferred based on the variability of the variables we have used so far in the Tropical Pacific. Note that the previous analysis is made for all seasons and without distinctions between certain types of events, say strong El-Niño events, as it is usually done when analyzing the effect of ENSO over other regions of the globe, e.g. Brönnimann (2007). Such a split between seasons and/or events are worth performing, but the time series are already short and the selection of certain events will reduce considerably the statistics. The absence of link between the Tropical Pacific and the North Atlantic coupled dynamics could also reflect the non-stationary properties of the teleconnections between the North Atlantic and the Tropical Pacific as documented in López-Parages et al (2016); Johnson and Kosaka (2016); Goss and Feldstein (2017) and references therein. The analysis of long climate runs of state-of-the-art models with the approach used here would be very useful in that respect.

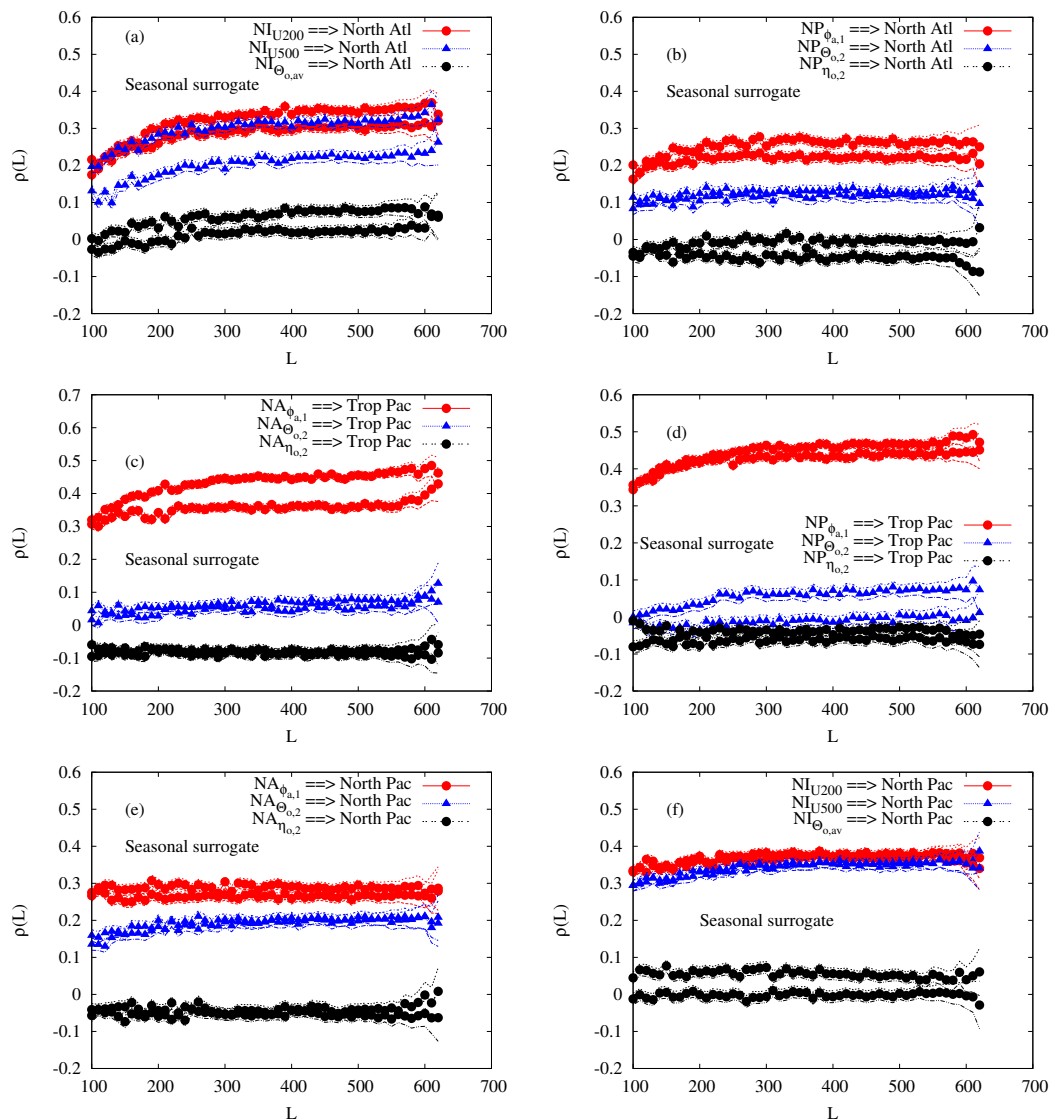

**Figure 5.** CCM as a function of the length L of the samples, as obtained from surrogate 3-dimensional attractors built by superimposing random anomalies to the annual climatological cycle, based on the data from Reanalysis ERA-20C and ORA-20C. Two different surrogates have been used. Each line with symbols corresponds to the influence of one actual variable (not a surrogate series) on a specific coupled ocean-atmosphere surrogate attractor. The specific variables are denoted in the caption corresponding to each line in each Panel.

## 4.2 ERA-20C/ORAS4

Let us now consider the second ocean reanalysis, ORAS4, while keeping the same atmospheric reanalysis. This second investigation should allow us to clarify the robustness of our findings. Figure 6 shows the results of the computation of CCM that should be compared with the results presented in Fig. 3. One remarkable feature is the absence of dependences between the North Atlantic and the North Pacific for the ocean dynamics, see black full circles in Panels (b) and (e). This result considerably differs from the one obtained with the ORA-20C. Another important difference is a larger amplitude of the influence of the Tropical Pacific ocean temperature on the North Pacific, and of the North Pacific variables on the North Atlantic. The other dependences are more robust.

These results suggest that the dynamics within the ocean differs considerably between ORAS4 and ORA-20C as already suggested by the covariances displayed in Table 1. These differences are probably due to (i) the fact that these reanalyses are obtained with different atmospheric forcing, specifically ORA-20C with ERA-20C and ORAS4 with different atmospheric reanalysis products and (ii) to the fact that for ORA-20C the ocean model started beginning of 1900 while for ORAS4 it started end of 1957. We may conjecture that the ORA-20C reanalysis data set is more reliable since a more consistent atmospheric forcing has been applied and the ocean model has got more time to equilibrate around its climate. But care should be taken here in drawing definitive conclusions on that. A better approach to disentangle which of these reanalyses provide the correct answer is to investigate a full coupled ocean-atmosphere reanalysis obtained for the whole 20th Century.

The investigation of the dependences for sliding averages over 12 months displayed at Fig. 7, also suggests a suppression of the dependences for most of the variables, at the exception of the one associated with the influence of the North Pacific ocean temperature on the Tropical Pacific (Panel (d)). It is also remarkable that a dependence emerges from the North Atlantic ocean dynamics to the North Pacific, but much weaker than for the other ocean reanalysis dataset.

Finally for the sake of completeness, the application of the CCM to the monthly anomalies is displayed in Fig. C2. Here as for the ERA-20C/ORA-20C dataset, no link between the anomalies of the Tropical Pacific and the North Atlantic is found (panels (a) and (c)). An important difference is however visible on the influence of the North Pacific ocean temperature on the North Atlantic and on the Tropical Pacific (panels (b) and (d)). Again, this contrasts with the results obtained with the other reanalysis dataset.

## 5 Conclusions

The causality between the dynamics of three different coupled ocean-atmosphere basins, The North Atlantic, the North Pacific and the Tropical Pacific region, NINO3.4, has been explored using data from three different reanalyses datasets, the ORA-20C, the ORAS4 and the ERA-20C. The approach used is the Convergent Cross Mapping developed by Sugihara et al (2012) which allows to go beyond the classical teleconnection patterns and which provides a clear signature of the inter-dependences between series or regions. The analysis reveals a few very important facts that should help improving our understanding of the remote influence of large-scale dynamical processes, and in particular the impact of the Tropical Pacific coupled dynamics on the extratropics:

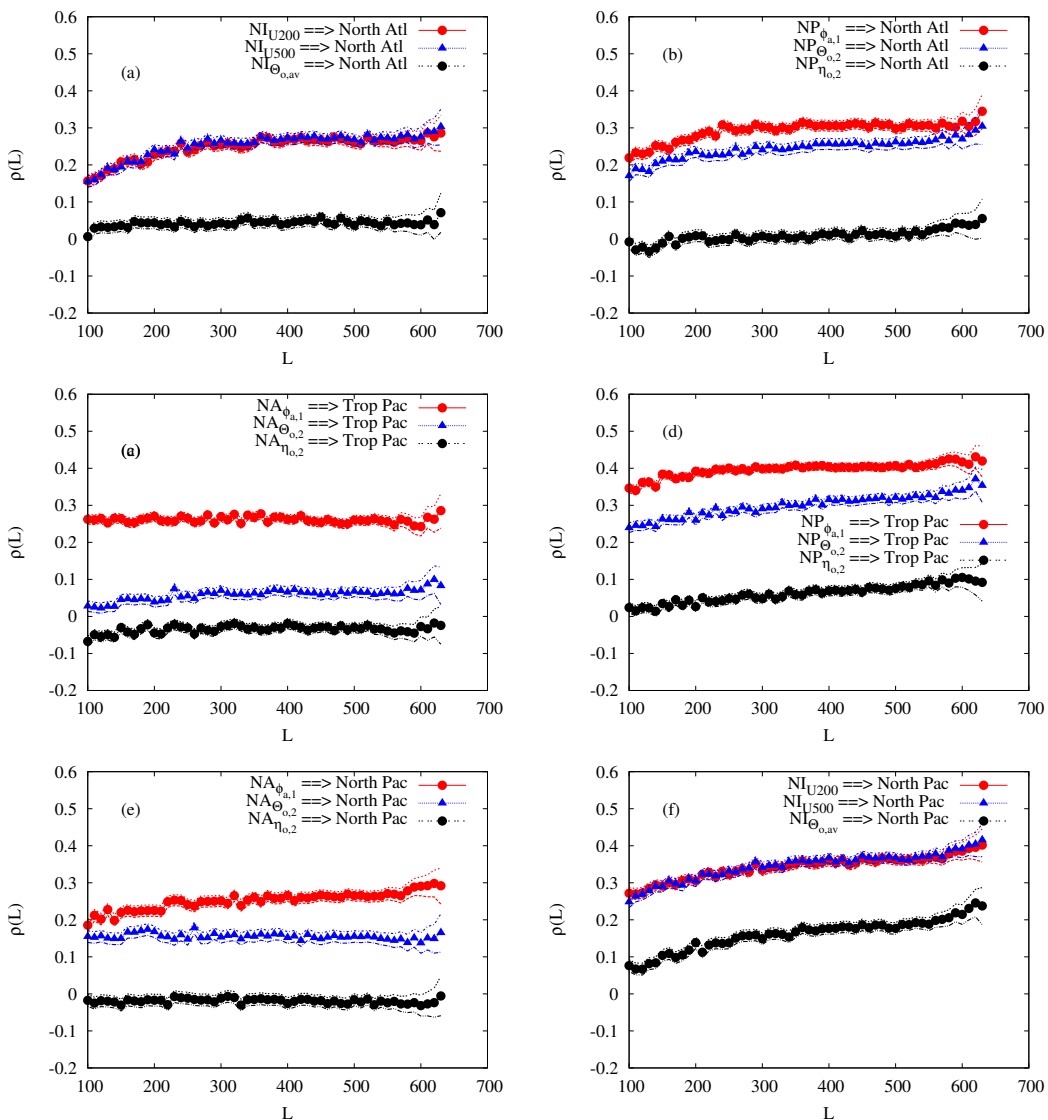

**Figure 6.** CCM as a function of the length L of the samples, as obtained from the monthly time series displayed in Fig. 2 for the reanalyses ERA-20C and ORAS4. Each line with symbols corresponds to the influence of one variable on a specific coupled ocean-atmosphere basin. The specific variables are denoted in the caption corresponding to each line in each Panel.

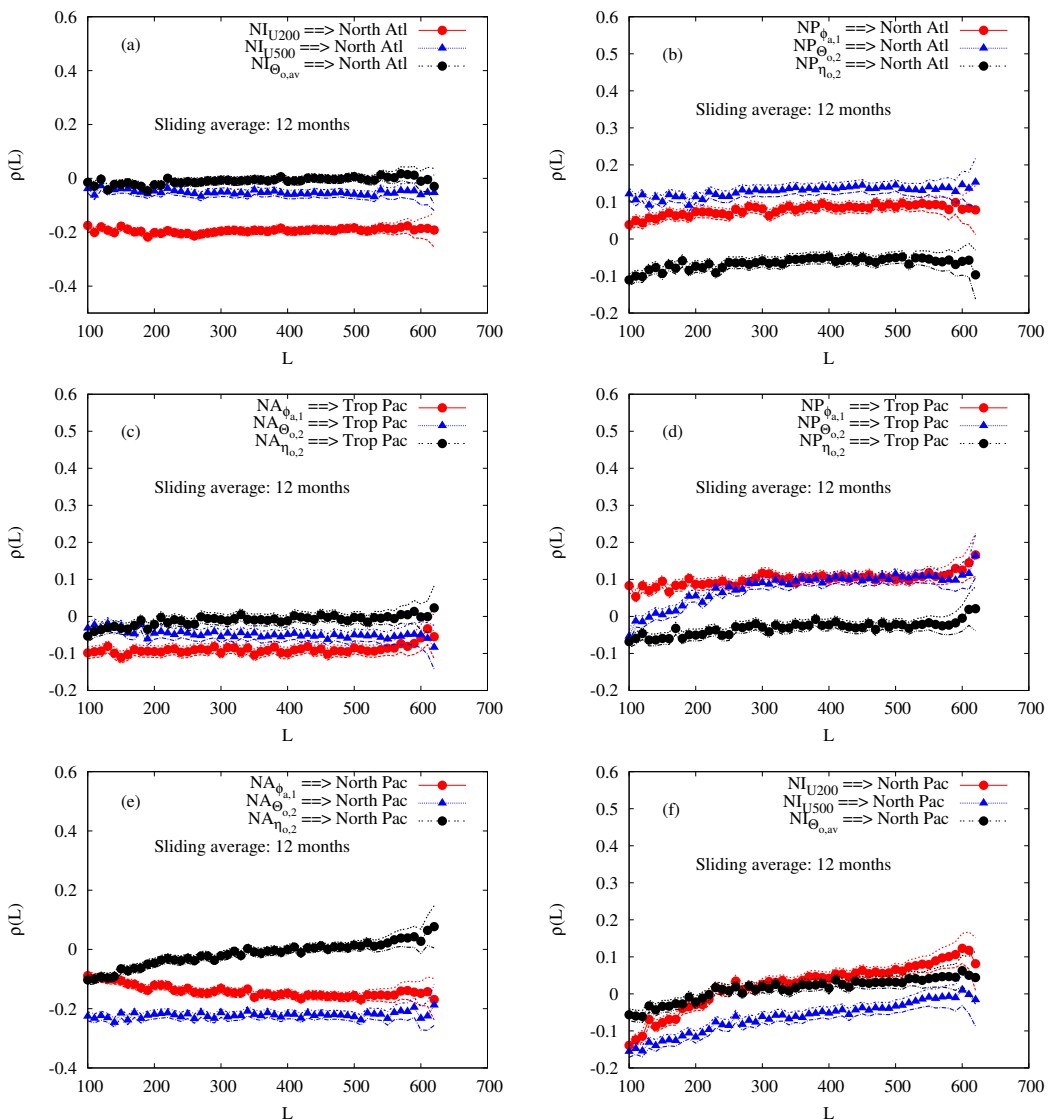

**Figure 7.** CCM as a function of the length L of the samples, as obtained from the series of Fig. 2 after averaging over a sliding window of 12 months for the reanalyses ERA-20C and ORAS4. Each line with symbols corresponds to the influence of one variable on a specific coupled ocean-atmosphere basin. The specific variables are denoted in the caption corresponding to each line in each Panel.

- The Tropical Pacific coupled ocean-atmosphere dynamics does not seem to have an impact on the extratropics beyond the annual climatological cycle. This very surprising result suggests that there is little hope to improve predictability in the extratropics based on information on the variability in the Tropical Pacific. This result needs however more attention and a thorough inspection of other datasets and long climate model runs in order to be confirmed. It is possible in particular that a more detailed analysis based on the selection of specific events, like strong El-niño or La-niña, will provide new information on the interactions between the Tropical Pacific and the rest of the world, beyond the climatological annual signal. But such an investigation needs considerably more data than the ones used in the present work and therefore calls for the development of even longer coupled reanalyses, or the use of long climate model runs.

- The atmosphere over the North Pacific considerably influences the North Atlantic (beyond the climatological annual signal), in agreement with the results found for instance in Drouard et al (2015).

- The results presented here for the two ocean reanalysis datasets disagree on the nature of the dependence between the North Pacific basin and the other ones. The ORA-20C indicates a dynamical influence (transport), while the ORAS4 suggests more an influence dominated by the ocean temperature. This difference is probably related to the specific data assimilation setup used in each case. To clarify what is really at work here new reanalysis datasets are needed. We can conjecture that a more reliable one could be built by using a coupled ocean-atmosphere data assimilation system running continuously on the longest period possible.

- The inter-dependences between the North Atlantic and the North Pacific on longer time scales than a year seems to be important, and is probably related to the coupled ocean-atmosphere dynamics on long time scales. One could conjecture that the thermohaline circulation should play a role on this link. Additional analyses with longer data sets, with climate model runs, but also with the analysis of additional basins like the Tropical Atlantic or the Indian Monsoon region are necessary to clarify this role.

The present work has demonstrated the urgent necessity to go beyond the teleconnection patterns for the investigation of the interaction between the different components of the climate system, using tools recently developed in the context of nonlinear sciences. Teleconnection patterns do provide information on a co-variability (which is an interesting information per se), but not of the influence of a region on another through a dynamical coupling. A common forcing can for instance induce a correlation between two variables even if these are perfectly independent in a dynamical sense.

New analyses will be performed along the lines drawn above, in particular in climate model runs. Several long control climate runs of CMIP5 models are available and can be analyzed in the same perspective as in the present work, in order in particular to evelute the impact of ENSO on the dynamics of the North Pacific and the North Atlantic.

Finally it should be stressed that the specific design of the CCM approach adopted here based on a low-dimensional projection of the full atmosphere-ocean attractor does not allow to have a one-to-one correspondence of the influence of one variable on another. Moreover other subspaces could be used that can provide different results. More variables should then be consid-

ered such as projections on additional Fourier modes, or by using projections on a few Empirical Orthogonal Functions. These analyses will allow to evaluate the robustness of the present results.

*Code and data availability.* The code for CCM and the time series are available upon request to the authors. The data are made available on zenodo.org.

## 5 Appendix A: The CCM algorithm

The different steps of the CCM algorithm used to compute the correlation coefficient is detailed in Fig. A1.

## Appendix B: CCM applied to an idealized model

To test the CCM technique described in Section 2, we use a dynamical system recently developed in Vannitsem (2015). It consists of a set of 36 ordinary differential equations representing the large scale dynamics of a coupled ocean-atmosphere
system at midlatitudes. The equations are described in Vannitsem (2015) and in its supplementary material. The atmospheric model is based on the vorticity equations of a two-layer, quasi-geostrophic flow defined on a $\beta$-plane. The ocean dynamics is based on the reduced-gravity, quasi-geostrophic shallow-water model on a $\beta$-plane. For the ocean, it is assumed that temperature is a passive scalar transported by the ocean currents, but the oceanic temperature field displays strong interactions with the atmospheric temperature through radiative and heat exchanges.
All fields are developed in Fourier series on a $\beta$-plane as,

$$\delta T_o = \sum_{i=2(and \neq 5)}^{8} \Theta_{o,i}\phi_i, \quad \Psi_o = \sum_{i=1}^{8} \Psi_{o,i}\phi_i, \tag{B1}$$

$$\psi_a = \sum_{i=1}^{10} \Psi_{a,i}F_i, \quad \delta T_a = \sum_{i=1}^{10} \Theta_{a,i}F_i \tag{B2}$$

$$\tag{B3}$$

where $\Theta_{a,i} = (\psi_{a,i}^1 - \psi_{a,i}^3)/2$ and $\Psi_{a,i} = (\psi_{a,i}^1 + \psi_{a,i}^3)/2$, with $\psi^1$ and $\psi^3$ the streamfunctions in the upper and lower layer
of the atmosphere. $\Psi_o$ is the streamfunction field in the ocean. $\delta T_o$ and $\delta T_a$ are temperature anomaly field with respect to spatially averaged reference temperatures. The modes $\phi_i$ used for the ocean are compatible with free-slip boundary conditions in a closed basin, while $F_i$ are modes used for the atmospheric fields compatible with free-slip boundaries in the meridional direction and periodic boundaries in the zonal direction, see also the paper Vannitsem et al (2015).

The CCM analysis is performed on the solutions generated by the model with the same parameters as in Fig. 3 of Vannitsem
(2015) with a surface friction coefficient $C = 0.006$ kg m$^{-2}$ s$^{-1}$. The model is forced with seasonal variations of the solar input as discussed in Vannitsem (2015), and the solutions are averaged over one month (1/12 of the 365 days of the model

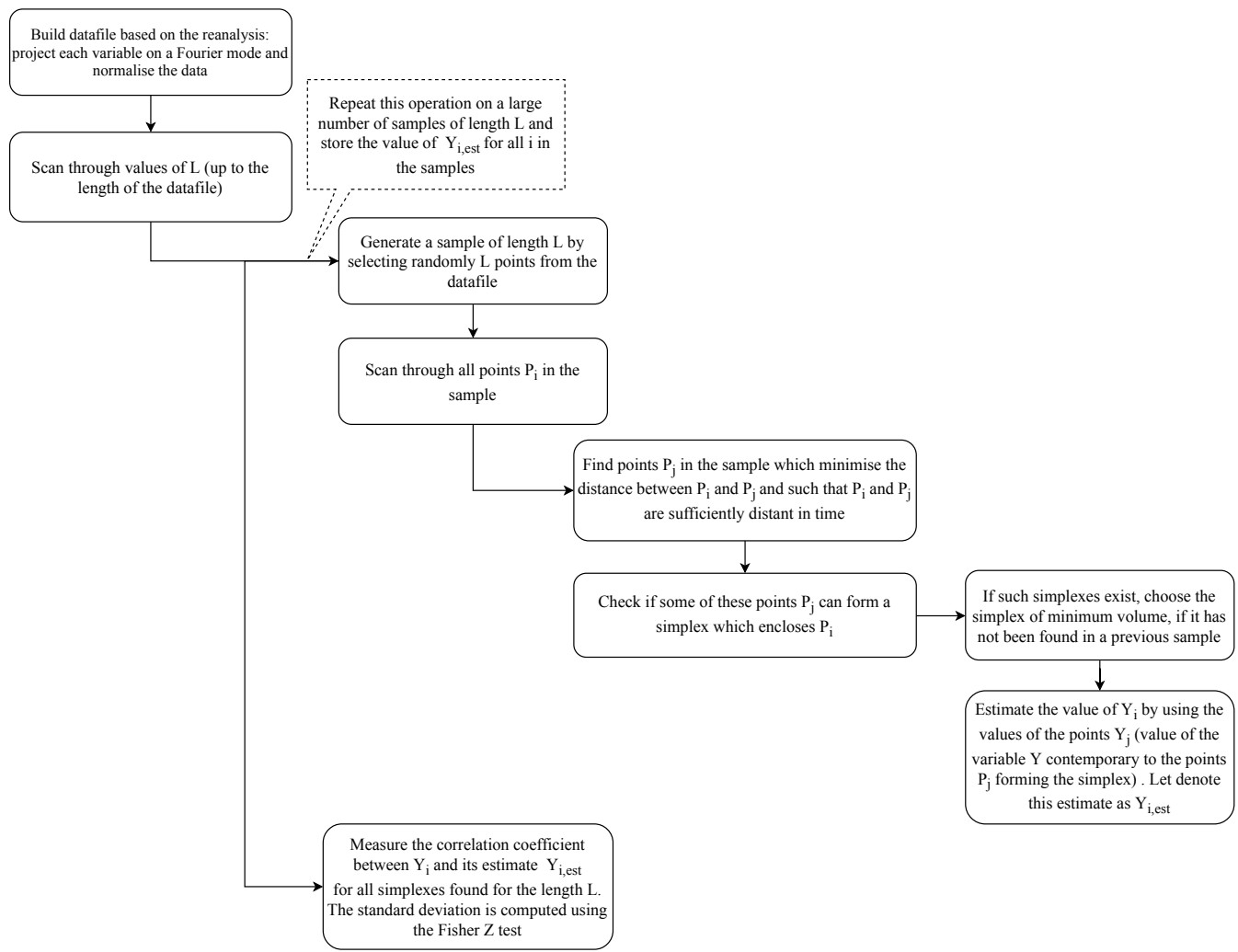

**Figure A1.** Algorithm used to compute the CCM presented in Section 2, based on the series discussed in Section 3.

year). The three variables used for building the attractor and the analogs are $(\Psi_{a,1}, \Theta_{o,2}, \Psi_{o,2})$ as for the North Atlantic and North Pacific datasets discussed in Section 3. Then several other variables are used to see if they have causality relations with the three variables used to build the attractor. The length of the time series is $L = 6000$ months.

The results are shown in Fig. B1 for different variables and also for a sliding window of 12 months at Panels (e) and (f). As it can be seen in Panels (a) and (b) the only atmospheric variable (explored so far) influencing the dynamics of $(\Psi_{a,1}, \Theta_{o,2}, \Psi_{o,2})$ is $\Theta_{a,1}$, since the correlation is high and increases as a function of $L$. This variable is strongly linked to the dynamics of $\Psi_{a,1}$ in the equations of the model, since it is the only one influencing (linearly) the evolution of $\Psi_{a,1}$. The others do not seem to have a strong influence. In Panels (c) and (d), the impact of some ocean variables is illustrated with no apparent influence of $\Psi_{o,5}$ and $\Theta_{o,3}$. But all other variables have different levels of influence with a clear increase of the correlation as a function of $L$.

Finally, in Panels (e) to (h) the impact of using a sliding average is illustrated. First the CCM plotted in Panel (f) indicates that the influence of $\Theta_{a,1}$ is not removed, indicating its essential role on the dynamics at different time scales of motion. Second the influence of the ocean modes which were found to play a role at monthly time scales is further enhanced.

This brief analysis of a low-order system based on the CCM algorithm described in Section 2 indicates that it is a powerful tool to isolate the influence of certain variables on others in the system. Note that the angle of approach adopted in Section 2 by considering a low-order projection of the full state space as target does not allow to have a detailed information on the nature of the coupling between the variables since what is inferred is a global influence of a variable on a subset of other variables. The analysis also opens new questions on the role of the different variables in this low-order model and the dependences on the specific subspace selected as the target. This problem is out of the scope of the present work but is worth pursuing in the future. We can now proceed with this approach in the context of the datasets presented in Section 3.

## Appendix C: Application of CCM to anomalies

The CCM is applied on the anomalies of the different time series displayed in Figs. 1 and 2. The anomalies are defined as $X(t,\tau) - \mu(\tau)$ where $\mu(\tau)$ is the average value over all years for month $\tau$, and the argument $t$ is the year for which the anomalies are computed. The analysis of these results is done in the core of the text.

*Author contributions.* TEXT

*Competing interests.* The authors declare no competing interest

*Disclaimer.* TEXT

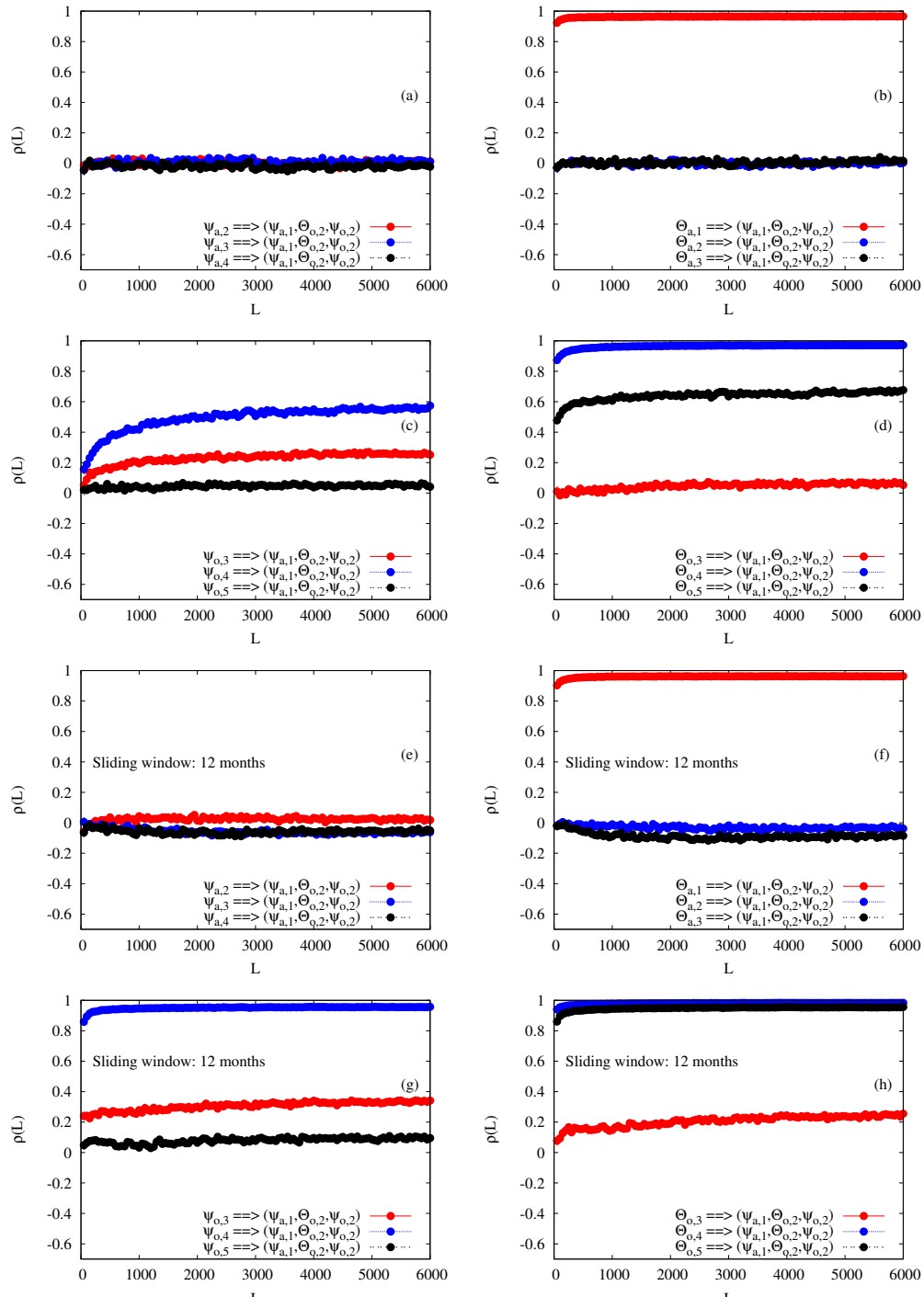

**Figure B1.** CCM as a function of the length L of the samples, as obtained from monthly time series of the low-order coupled ocean-atmosphere model integration. Panels (a) to (d) displays the values for the influence of a set of model variables on $(\Psi_{a,1}, \Theta_{o,2}, \Psi_{o,2})$ at monthly time scale. Panels (a) to (d) displays the values for the influence of a set of model variables on $(\Psi_{a,1}, \Theta_{o,2}, \Psi_{o,2})$ after the application of a sliding average over 12 months. Each line with symbols corresponds to the influence of one variable on $(\Psi_{a,1}, \Theta_{o,2}, \Psi_{o,2})$. The specific variables are denoted in the caption corresponding to each line in each Panel.

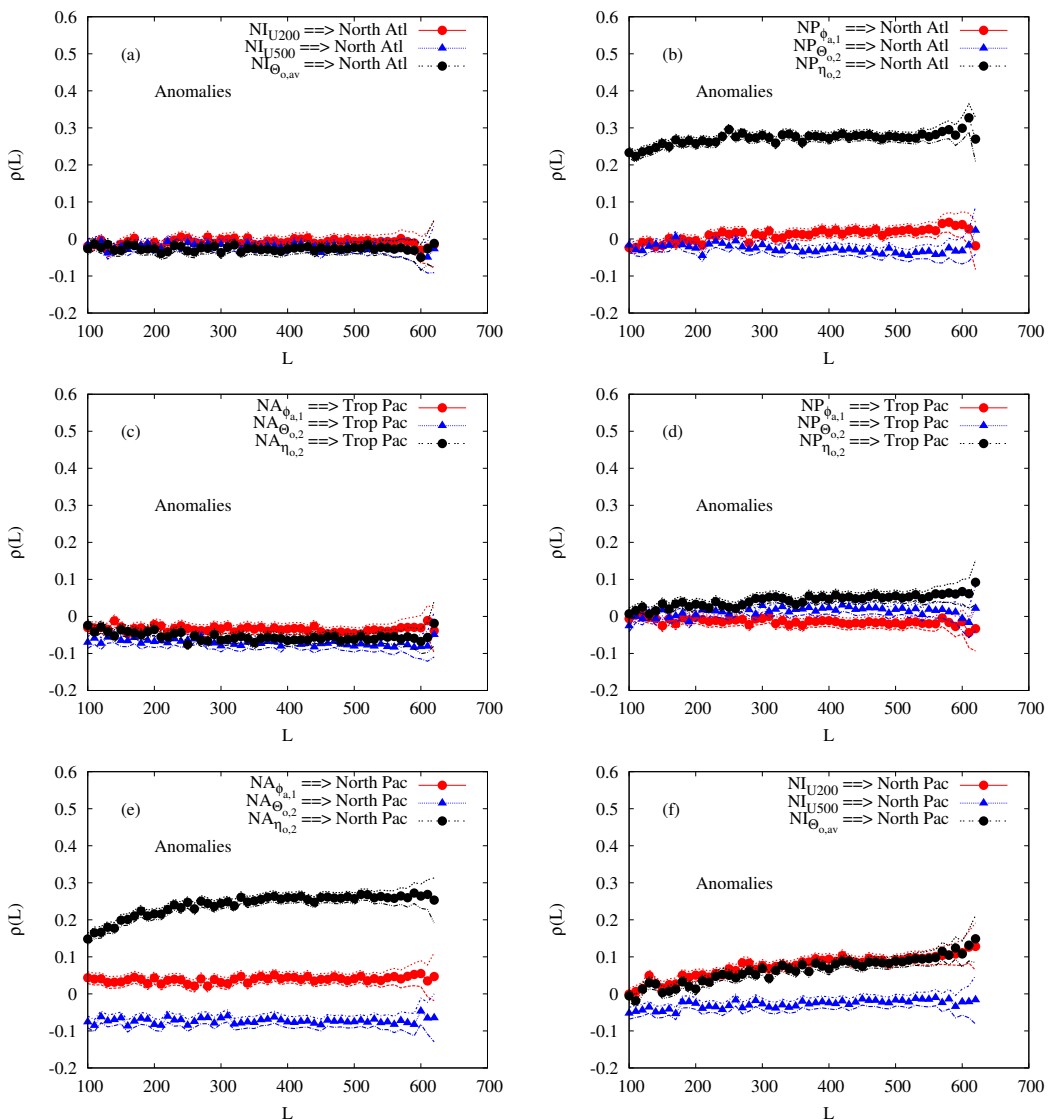

**Figure C1.** CCM as a function of the size $L$ of the samples, as obtained from the anomaly of the monthly time series displayed in Fig. 1 for the reanalyses ERA-20C and ORA-20C. Each line with symbols corresponds to the influence of one variable on a specific coupled ocean-atmosphere basin. The specific variables are denoted in the caption corresponding to each line in each Panel.

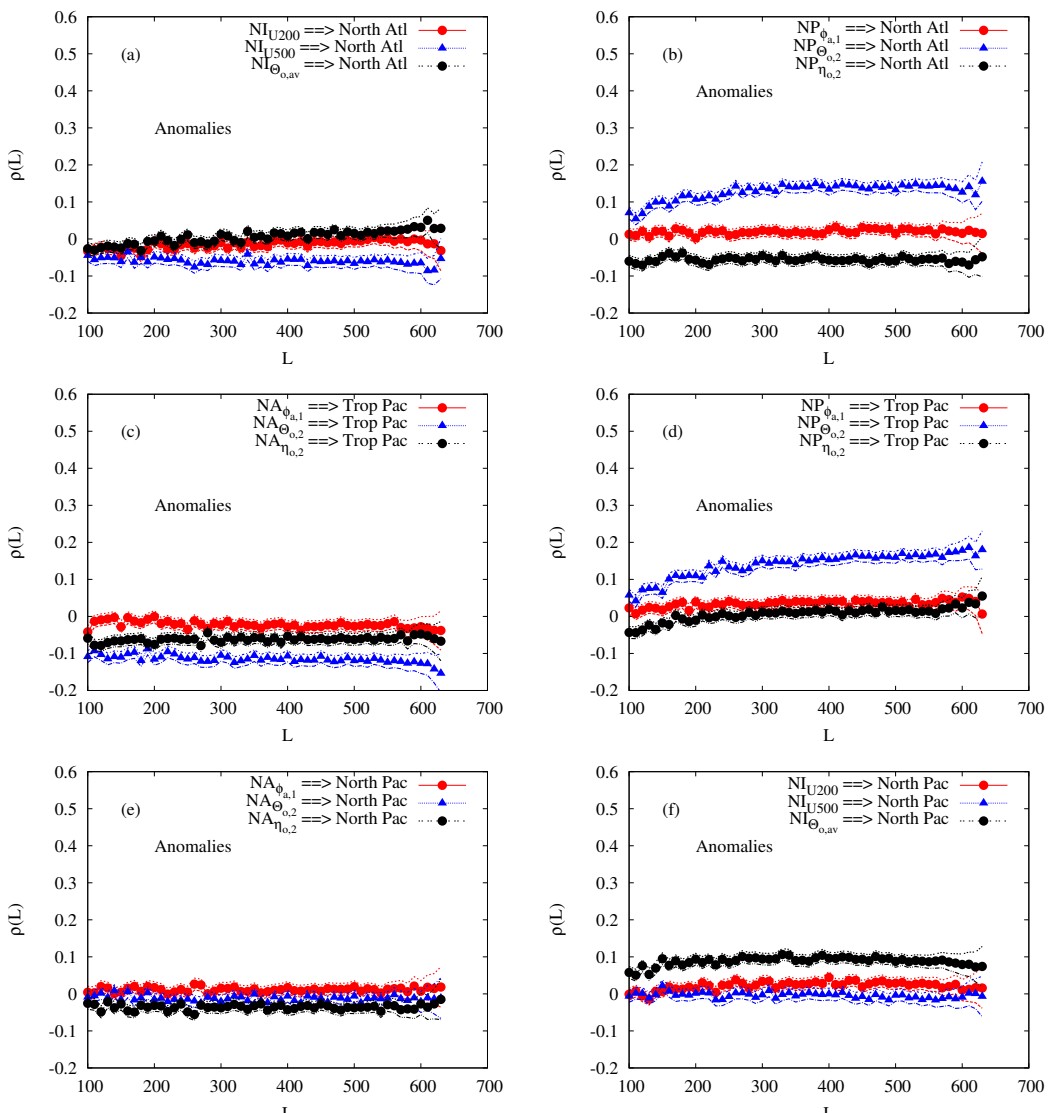

**Figure C2.** CCM as a function of the size $L$ of the samples, as obtained from the anomaly of the monthly time series displayed in Fig. 2 for the reanalyses ERA-20C and ORAS4. Each line with symbols corresponds to the influence of one variable on a specific coupled ocean-atmosphere basin. The specific variables are denoted in the caption corresponding to each line in each Panel.

*Acknowledgements.* This work is supported in part by the EU project MEDSCOPE (ERA4CS) and the project Mass2Ant of the Belgian Scientific Policy Office.

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
