# Peer review of "Causal dependences between the coupled ocean-atmosphere dynamics over the Tropical Pacific, the North Pacific and the North Atlantic"

_Earth System Dynamics, 2018_

## Referee Comment (RC1) · Anonymous Referee #1 · 27 Jan 2018

In this manuscript the authors investigate causal relationships, in monthly to inter-annual time scales, between the climate dynamics of three ocean-atmosphere basins (The North Atlantic, the North Pacific and the Tropical Pacific region) using three re-analyses datasets (ORA-20C, ORAS4 and ERA-20C). The applied methodology, Convergent Cross Mapping (CCM) has been applied to other systems, but has not yet been used (to the best of my knowledge) to investigate climate causal relationships. I found this study very interesting. As the authors acknowledge in the introduction, unveiling causal relations is a very challenging task, and different methods (also depending on

the choice of variables), are likely to yield different results. Here the CCM method is well motivated and described, and also the datasets used for reconstructing three-dimensional attractors are well justified. The results obtained are sound. Therefore, I am happy to recommend the acceptance of this manuscript, after the authors have taken the following points.

1) By using three time series, the authors reconstruct three-dimensional attractors (instead of using one time series and Taken's delayed coordinates). Could the authors discuss how important is the method used to reconstruct the attractor and the chosen attractor dimension? What could be expected if i) two-dimensional attractors are reconstructed from two time series (instead of three, using, e.g., only the zonal velocity at either 200 or 500 hPa and the ocean temperature)? and ii) three-dimensional attractors are reconstructed from one time series using Taken's delayed coordinates?

2) The authors state that "If there is a causality relation of Y on X, "ro" (Eq. 3) will increase with L". However, what this study uses (and I am not sure is always true) is the fact that "an increase of "ro" with L reveals/uncovers a causal relation of Y on X". Could the authors discuss the limits of validity of these two statements? I assume they hold if "L" is appropriated (not too small, and not too close to the length of the dataset). How about if X and Y are both driven by Z?

Minor comments are

3) In the Introduction, the authors say "an important question nowadays is to know whether the Tropical Pacific system forces the dynamics of the climate system in the extratropics". In my view there is plenty of evidence (it is well known) that the Tropical Pacific system forces the dynamics of the climate system in the extratropics, and therefore, I suggest the authors to re-word this sentence.

4) Regarding the link between galactic cosmic rays and global temperature variations, there is a discussion (Questionable dynamical evidence for causality between galactic cosmic rays and interannual variation in global temperature, doi:

10.1073/pnas.1510571112 and the reply, DOI: 10.1073/pnas.1511080112) that the authors, in my view, should also cite.

5) Table 1 would be easier to read if there is a space between the numbers (i.e., instead of x|y, x | y), also the letters in the figures are too small.

---

## Referee Comment (RC2) · Anonymous Referee #2 · 11 Mar 2018

The paper discusses the use of a relatively new method to infer causal dependencies in climate dynamics. While the subject is of profound interest I am not convinced about the contributions of this paper. In short, the method is introduced as an ad hoc method, its description is confusing, and the results cannot be reprodueced as crucial information is missing. Also, the conclusions are not fully backed up by the results. I suggest either a major revision or rejection, based on arguments detailed below.

1. There is a quickly growing literature on causal inference methods based in information theory. The authors provide one sentence on page 2 lines 21-23 on it, saying

[Figure]

something like 'finding a good estimator for analyzing real data is difficult'. I have no idea what this is about, and see no reason why the information theoretic methods could not be used here, even with the time series used in this paper. Please provide a proper argument on why these methods cannot be used, and why the proposed CCm method is better/more suitable. My fear is there is no such argument.

2. section 2: The authors talk about phase space while they mean state sspace. A phase space is a position-velocity space, which is of interest when describing the motion of particles, but not for fluid dynamics, which is governed by first-order time derivatives. Please correct.

3. The distance measure is a crucial ingredient in CCM as far as I understand it, yet no discussion is provided. Saying that typically the Euclidian distance is used is not useful, more discussion is needed. This is related to the next point.

4. The weights in eq 2 seem to be completely arbitrary. First I assume that the authors use something like exp(-d_j/dmin), otherwise the whole procedure doesn't seem to make sense. But that might be just a typo. More importantly is why this is a good weighting, also relation to Euclidian distances. As far as I can see this is completely ad hoc. At least a justification for this choice of weights needs to be given. Furthermore, the question immediately arises how sensitive the results are to form of the weights. This is not discussed. Please do!

5. The prediction defined by eq 1 depends on which analogs have been used, so on the prior sample. How large is the sensitivity of the results to that?

6. The CCM methods starts out strong with the embedding, but then falls back to correlations, with all weaknesses that the paper wants to avoid in the first place. For instance, if X(t) = sin(omega t) and y(t) = X(t - pi/(2 omega)) the relation between X and Y is circular, so the correlation in eq 3 is zero. This, however, would negate the existence of a very strong causal relation.

7. As far as I can see, if the causal relation between X and Y peaks strongly at say 2 tau adding more elements in X will not increase the correlation, it might even make the correlation smaller. But the authors say differently in e.g. p4 lines 19-25. Please clarify.

8. section 3. The authors assume that the main ocean dynamics is governed by Sverdrup balance in mid latitudes. This might be true for the large-scale dynamics, but is not so for scales smaller than a few 100 km. Are the main conclusions of the paper justified when all dynamics of the ocean beyond Sverdrup are ignored? For instance, isn't is essential for the North Atlantic circulation and the interaction with the atmosphere, e.g. via the storm tracks, that the GulfStream extension is not zonal? I think the authors should at least justify much more strongly why they think they can use this simple mode structure to project the reanalyses data sets on.

9. The authors use the monthly fields for the analysis. Are these snapshots or monthly averages?

10. What was dmin for each time series?

11. The authors calculate a CI, and there is a growing tendency in the science community to eradicate its use because of its arbitrariness, among bother problems. Can the authors instead provide error estimates on their results?

12. It is important that the analogs are independent from each other. How did the authors ensure that? Please at least discuss this.

13. Please make the legends of all figures larger, and the error estimate lines thicker.

14. Fig 3 $^{©}$ and (e) should be interchanged(?)

15. What happens for low time lags in fig 4f?

16. Please describe the distribution of the random anomalies in page 13 line 9.

17. page 13 lines 18-24. I don't necessarily agree with the authors' conclusion on the fact that the annual surrogates give the same correlations as the full time series. This

does not mean that there is no underlying dynamical causal structure, only that the annual signal is dominant. One has to be careful, as the annual signal is very, very strong. While the authors argue that they don't want to try to remove the annual signal from the time series because of the issues with doing that, perhaps they have to do it this time, e.g. by using different methods and compare results.

18. section 4.2. I'm not sure one can say that ORA-20C is more reliable than ORAS4. I understand from their description that both do not assimilate ocean variables, they only differ in their atmospheric forces. But the ocean circulation is not only determined by the surface forcing, but also, and perhaps to a much larger extend, on the initial condition and on model biases. How do the initial conditions differ between the two ocean products, and assuming they are both initialised from a smoothed observational data set of T and S observations, could it be the case that the ORA-20C run has a less realistic ocean vertical structure because ocean model biases have degraded that longer compared to ORAS4?

19. The conclusions are too strong given the CCM method used, the high simplification of the Earth system dynamics to a few in my view quite unrealistic modes, apart from the other potential problems mentioned above. The first conclusion especially, see point 17 above, but also the 4th conclusion. The authors seem to miss the possibility that the atmospheric forcing needs some time to give visible changes in the ocean, e.g. the gyre circulation needs a few years to adapt to a change in the wind stress. It is not necessarily ocean only dynamics that sets time scales of over a year.

20. The fact that the CCM method is useful for the simplified model in the appendix does not provide a trong argument that it will work for the real ocean as the simplified model does indeed only have a few modes, while the real Earth system has many, with interactions that are much more complicated and more nonlinear. Please provide a stronger justification on why this experiment backs up the analysis on the far more complicated real system.

---

## Author Response (AR1)

**Response to the comments of Reviewer 1, Reviewer 2 and the editor**

May 11, 2018

**Stéphane Vannitsem**[1] **and Pierre Ekelmans**[2]

[1]Royal Meteorological Institute of Belgium, Avenue Circulaire, 3, 1180 Brussels, Belgium

[2]Theory of Neural Dynamics Group, Max Planck Institute for Brain Research, Max-von-Laue-Strasse 4, 60438 Frankfurt, Germany

This document provides the detailed answers to the remarks and questions of the reviewers and the editor. The points of the Reviewer are italicized, and the (proposed) modifications in the text are in bold face. The pages and lines refer to the manuscript attached along with the response.

**1 Reviewer 1**

*In this manuscript the authors investigate causal relationships, in monthly to inter-annual time scales, between the climate dynamics of three ocean-atmosphere basins (The North Atlantic, the North Pacific and the Tropical Pacific region) using three reanalyses datasets (ORA-20C, ORAS4 and ERA-20C). The applied methodology, Convergent Cross Mapping (CCM) has been applied to other systems, but has not yet been used (to the best of my knowledge) to investigate climate causal relationships. I found this study very interesting. As the authors acknowledge in the introduction, unveiling causal relations is a very challenging task, and different methods (also depending on the choice of variables), are likely to yield different results. Here the CCM method is well motivated and described, and also the datasets used for reconstructing three- dimensional attractors are well justified. The results obtained are sound. Therefore, I am happy to recommend the acceptance of this manuscript, after the authors have taken the following points.*

We would like first to thank the reviewer for her/his positive support to this work. Below we answer her/his specific points.

*1 By using three time series, the authors reconstruct three-dimensional attractors (instead of using one time series and Takens' delayed coordinates). Could the*

*authors discuss how important is the method used to reconstruct the attractor and the chosen attractor dimension? What could be expected if (i) two-dimensional attractors are reconstructed from two time series (instead of three, using, e.g., only the zonal velocity at either 200 or 500 hPa and the ocean temperature)? and (ii) three-dimensional attractors are reconstructed from one time series using Takens delayed coordinates?*

The motivation of this approach is to avoid the difficulties associated with the embedding and the delay. Now the choice of three variables for the Atlantic and Pacific is motivated by the results we have obtained in previous works on the low-dimensional modelling of the coupled ocean atmosphere system, in which we have found that these three variables dominates the dynamics (Vannitsem et al, 2015; Vannitsem, 2015). This 3D space constitutes a kind of projection of the full phase space. For the Tropical Pacific, the choice is motivated by the importance of the surface temperature and the zonal wind in the development of the coupled ocean-atmosphere dynamics. For the latter region we could indeed imagine to use only 2D maps.

(i) Using 2D maps, useful results can also be obtained. An example is given here for the influence of the Tropical time series on the North Atlantic in Figure 1. In the different results presented in this figure, the only influence which emerges is from the atmospheric Tropical observables to the 2D map defined by the Geopotential at 500 hPa and the Ocean Temperature. But the growth as a function of $L$ is weaker than the one obtained when using the 3D map (Fig 3a of the manuscript). As mentioned by Sugihara et al (2012), the variation of $\rho$ with $L$ is faster before saturation when the strenght of coupling is higher. Here the coupling between the variables when a 2D map is used seems to be less important than when using the three variables, suggesting that the Tropical Pacific atmosphere indeed influence the coupled ocean-atmosphere system as defined by the three observables. In other words, the latter result suggests that the dependences between these regions is better elucidated based on the three dimensional space. Care should however be taken here drawing definitive conclusions on this comparison since the setup of the analysis has considerably changed with a change of phase space dimensionality and a reduction of the number of analogs used (only three in this case).

In the text, we add at page 11, line 19:
**Note that the reduction of the state space coordinates associated with $\vec{X}$ from three to two also provides interesting results with a dominating influence from the atmospheric Tropical Pacific variables on the two-dimensional projection $(NA_{\Psi_{a,1}}, NA_{\theta_{o,2}})$. However the increase before saturation of $\rho(L)$ is much more limited than when using the three variables to build the North Atlantic projection of the attractor (not shown). The latter result suggests that the dependences between these regions is better elucidated based on the three dimensional space.**

(ii) When building attractors using the Takens' theorem, one needs to define an embedding dimension and a delay $\tau$. Estimating the embedding dimension based for instance on the estimates of the correlation dimension of the attractor is very challenging when the expected embedding dimension is high since the approach needs to select close analogs to work properly (e.g. Kantz and Schreiber, 1995).

[Figure]

Figure 1: Example of CCM as a function of the length $L$ of the samples, as obtained from the 2D maps for the reanalyses ERA-20C and ORA-20C. The influence investigated here is the one from the Tropical Pacific on the North Atlantic. Each line with symbols corresponds to the influence of one variable on a specific coupled ocean-atmosphere basin. The specific variables are denoted in the caption corresponding to each line in each Panel. The 2D maps used are (a) Geopotential at 500 hPa + Ocean Streamfunction, (b) the Geopotential at 500 hPa + Ocean Temperature and (c) the Ocean Temperature + Ocean Streamfunction.

It therefore needs very long time series that are usually not affordable (Van den Dool, 1994, Nicolis, 1998). So a way to overcome this problem is to increase progressively the embedding dimension and see whether the results are robust or not. For the delay $\tau$, one usually uses a time period for which successive situations become sufficiently decorrelated, but not too much. Different methods are usually proposed to evaluate this delay, for instance based on decorrelation times, or simply by trial and error (e.g. Casdagli, 1991; Parker and Chua, 1989). In the present cases these delays should be relatively short for the atmosphere, but much longer for the ocean as it can be guessed by inspecting the time series of the right panels of Figures 1 and 2 of the manuscript. For the latter we are therefore facing an important problem since the decorrelation time (or delay) is not substantially smaller than the length of the time series. In the present context, we therefore opt for another approach based on selecting a subset of variables, a projection on a low-dimensional space.

We have modified the text after line 14 at page 5 to clarify point (ii):

**Estimating the embedding dimension based for instance on the estimates of the correlation dimension is very challenging when the expected dimension is high since the approach needs to select close analogs to work properly (e.g. Kantz and Schreiber, 1995). It therefore needs very long time series that are usually not affordable (Van den Dool, 1994, Nicolis, 1998). So a way to overcome this problem is to increase progressively the embedding dimension and see whether the results are robust or not. For the delay $\tau$, one usually uses a time period for which successive situations become sufficiently decorrelated, but not too much. Different methods are usually proposed to evaluate this delay, for instance based on decorrelation times, or simply by trial and error (e.g. Casdagli, 1991; Parker and Chua, 1989). In the present cases these delays should be relatively short for the atmosphere, but much longer for the ocean as it can be guessed by inspecting the time series of the right panels of Figures 1 and 2. For the latter we are therefore facing an important problem since the decorrelation time (or delay) is not substantially smaller than the length of the time series.**

We also added a sentence explaining what information can be extracted from the growth of $\rho(L)$ at page 5, line 3.

**Another important behavior of $\rho(L)$ as a function of $L$ is that the rate of increase is related with the strength of coupling.**

*2) The authors state that "If there is a causality relation of Y on X, "ro" (Eq. 3) will increase with L". However, what this study uses (and I am not sure is always true) is the fact that "an increase of "ro" with L reveals/uncovers a causal relation of Y on X". Could the authors discuss the limits of validity of these two statements? I assume they hold if "L" is appropriated (not too small, and not too close to the length of the dataset). How about if X and Y are both driven by Z?*

This point is one key element of the method. The main hypothesis behind these statements is that when extra information (larger L) is present, then one should expect to get better analogs around X and therefore to get better knowledge on the correct value of Y. This has been amply demonstrated on different simple systems

by Sugihara et al (2012), and also by other authors. Note also that in an ideal context where the attractor can be reconstructed with precision and for $L$ going to infinity, the correlation should converge to 1. In practical situations, this precision and the asymptotic limit are never reached. The convergence is then limited to a certain level by the presence of observational error, the approximation of the dynamics (like when a low-dimensional approximation is made of the full system) and the length of the series, $L$.

When a common driver Z is at play on X and Y, and that X and Y are independent, then the correlation between $Y$ and $\hat{Y}$ is positive but it does not depend on the data set length $L$. So a constant value as a function of L is expected. This has also been shown in Sugihara et al (2012). So we added at line 6, page 5:

**For instance, if a confounding factor $Z$ affects both $\vec{X}$ and Y (that are otherwise independent of each other), they will contain a similar information, and the inference of $\hat{Y}$ based on $\vec{X}$ will display a correlation with $Y$, but which is independent of $L$ (Sugihara et al, 2012).**

To explain the behavior as a function of $L$, we also add at page 5, line 9:
**Note also that in an ideal context where the attractor can be reconstructed with precision and for $L$ going to infinity, the correlation should converge to 1. In practical situations, this precision and the asymptotic limit are never reached. The convergence is then limited to a certain level by the presence of observational error, the approximation of the dynamics (like when a low-dimensional approximation is made of the full system) and the length of the series, $L$.**

*Minor comments are*
*3) In the Introduction, the authors say "an important question nowadays is to know whether the Tropical Pacific system forces the dynamics of the climate system in the extratropics". In my view there is plenty of evidence (it is well known) that the Tropical Pacific system forces the dynamics of the climate system in the extratropics, and therefore, I suggest the authors to re-word this sentence.*

Right. We reformulate the sentence as follows (page 2, line 4):
**In particular, an important question that has attracted a lot of attention in the past decades is to know whether the Tropical Pacific system *forces* the dynamics of the climate system in the extratropics.**

*4) Regarding the link between galactic cosmic rays and global temperature variations, there is a discussion (Questionable dynamical evidence for causality between galactic cosmic rays and interannual variation in global temperature, doi: 10.1073/pnas.1510571112 and the reply, DOI: 10.1073/pnas.1511080112) that the authors, in my view, should also cite.*

Thank you very much for these interesting references. These are incorporated in the new version of the manuscript.

*5) Table 1 would be easier to read if there is a space between the numbers (i.e., instead of x—y, x — y), also the letters in the figures are too small.*

Thank you very much for the suggestion. We did it.

**2    Reviewer 2**

*1. There is a quickly growing literature on causal inference methods based in information theory. The authors provide one sentence on page 2 lines 21-23 on it, saying something like "finding a good estimator for analyzing real data is difficult". I have no idea what this is about, and see no reason why the information theoretic methods could not be used here, even with the time series used in this paper. Please provide a proper argument on why these methods cannot be used, and why the proposed CCm method is better/more suitable. My fear is there is no such argument.*

One of the present authors used a technique developed by Liang (2014) to compute the dependences based on information flow. This technique has been built assuming that the system is linear, and Liang (2014) found experimentaly that this approach could work for nonlinear systems as well. The estimator in this case is a combination of correlations and derivatives of correlations. This approach has been extended by introducing a normalization (Liang, 2015). One problem pointed out by the Liang (2014) is the fact that several choices can be made for the discretization of the derivative. Another aspect is that one cannot have dependences without correlation. The latter situation seems surprising since some systems (like the one pointed out by the reviewer in his/her remark 6) can display dependences without correlation. In view of these difficulties, we decided to postpone the investigation based on these approaches to a future work.

We understand that our comment in the mansucript is not based on sufficiently firm grounds, so we modified it in the following way (page 2, line 28):

**Alternative approaches based on the transfer of information are very appealing and a lot of progresses have been made in that direction (Liang and Kleeman, 2005, Runge t al, 2012, Liang, 2014, Liang, 2015). We however do not pursue in that direction and let the use of these techniques for a follow-up study.**

*2. section 2: The authors talk about phase space while they mean state space. A phase space is a position-velocity space, which is of interest when describing the motion of particles, but not for fluid dynamics, which is governed by first-order time derivatives. Please correct.*

Well the phase space is a terminology which is not only valid for position-velocity space. It is used in general to describe the abstract space spanned by the coordinates which are the variables of the dynamical system themselves, and it is used interchangeably with the other terminology of "state space". I refer to classical books like Nicolis (1995) or Broer and Takens (2011). We do not see why we have to change the terminology, but we modified it under the insistence of the Editor.

*3. The distance measure is a crucial ingredient in CCM as far as I understand it, yet no discussion is provided. Saying that typically the Euclidian distance is used is not useful, more discussion is needed. This is related to the next point.*

We have modified the sentence by expliciting the distance used. Other distances can be used. The text added at page 3, line 33:

$d_i = \sqrt{\sum_j (X_j(t) - X_{j,i}(t))^2}$ **where $X_j(t)$ and $X_{j,i}(t)$ are the (delay) coordinates of the reference point and the ith analog, respectively. Other distances could be used.**

*4. The weights in eq 2 seem to be completely arbitrary. First I assume that the authors use something like $exp(-d_j/d_m in)$, otherwise the whole procedure doesn't seem to make sense. But that might be just a typo. More importantly is why this is a good weighting, also relation to Euclidian distances. As far as I can see this is completely ad hoc. At least a justification for this choice of weights needs to be given. Furthermore, the question immediately arises how sensitive the results are to form of the weights. This is not discussed. Please do!*

You are right. Thank you very much for pointing out that. Now the weight given in Eq 2 is modified.

The development of this type of nonlinear forecasting traces back to several seminal papers, see for instance (Casdagli, 1991, Elsner and Tsonis, 1992) for a detailed discussion. In its original version, more general weights $w_i$ were proposed that should be fitted through a least square approach (Casdagli, 1991). Sugihara and May (1990) proposed to simplify the approach by using a simpler variant based on exponential functions depending on the distance between the analogs and the reference point. This weighting penalizes analogs that are far from the reference point, and the normalization by the minimum distance allows for having weights based only on the relative distance. This technique works well as discussed in Sugihara and May (1990). Moreover it does not need any additional parameter, implying that the approach is parcimonious.

We have added in the manuscript at page 4, line 11:

**The development of this type of nonlinear forecasting traces back to several seminal papers, see for instance Casdagli (1991) and Elsner and Tsonis (1992) for a detailed discussion. In their original versions, more general weights $w_i$ were proposed that should be fitted through a least square approach (Casdagli, 1991). Sugihara and May (1990) proposed to simplify the approach by using a simpler variant based on exponential functions depending on the distance between the analogs and the reference point. This weighting penalizes analogs that are far from the reference point, and the normalization by the minimum distance allows for having weights based only on the relative distance. This technique works well as discussed in Sugihara and May (1990). Moreover it does not need any additional parameter, implying that the approach is parcimonious.**

*5. The prediction defined by eq 1 depends on which analogs have been used, so on the prior sample. How large is the sensitivity of the results to that?*

This is true, and in order to evaluate the uncertainty associated with the choice of the sample $L$, we have performed a large number of other random sampling and provide the mean value and a standard deviation as a measure of uncertainty. We added a paragraph on that at the end of Section 2 (page 6, line 9):

**Note that in order to evaluate the impact of the random sampling of $L$ events in the datasets, we can repeat the sampling a certain number**

of times and infer a mean (or a median when strong assymetries are present) and a standard deviation. In the experiments that will be described below, this approach is adopted and each correlation value is estimated over a large number of samples.

*6. The CCM methods starts out strong with the embedding, but then falls back to correlations, with all weaknesses that the paper wants to avoid in the first place. For instance, if X(t) = sin(omega t) and y(t) = X(t - pi/(2 omega)) the relation between X and Y is circular, so the correlation in eq 3 is zero. This, however, would negate the existence of a very strong causal relation.*

It seems that the reviewer is confusing the correlation between $X$ and $Y$, and the correlation between $\hat{Y}$ and $Y$, where $\hat{Y}$ is the estimate based on the $X$ attractor. This correlation is a measure of the quality of inference which has been used for a long time now, and which is still used to evaluate weather forecasts. This measure is related to the error between the inferred situation and the observed one.

Coming back to the example given by the reviewer, what is matter is the inference $\hat{Y}$ based on $X$ and what we get with this circular solution is a correlation between $\hat{Y}$ and $Y$ very close to 1 (and not equal to 0). So a kind of perfect inference. Now there is no increase as a function of $L$ due to the fact the same source of information is present in both, the signal $X$.

*7. As far as I can see, if the causal relation between X and Y peaks strongly at say 2 tau adding more elements in X will not increase the correlation, it might even make the correlation smaller. But the authors say differently in e.g. p4 lines 19-25. Please clarify.*

Well it seems that the reviewer is confusing the approach based on delay embedding and the approach adopted here in which contemporary data are used. We add the term **contemporary** in the description of the approach at page 5, line 24.

*8. section 3. The authors assume that the main ocean dynamics is governed by Sverdrup balance in mid latitudes. This might be true for the large-scale dynamics, but is not so for scales smaller than a few 100 km. Are the main conclusions of the paper justified when all dynamics of the ocean beyond Sverdrup are ignored? For instance, isn't is essential for the North Atlantic circulation and the interaction with the atmosphere, e.g. via the storm tracks, that the GulfStream extension is not zonal? I think the authors should at least justify much more strongly why they think they can use this simple mode structure to project the reanalyses data sets on.*

Thank you very much for the comment. Indeed the projection used here is meant to investigate the large scale dynamics and the link between the different ocean basins. There is by no means any aim at providing or using information at smaller scales. We add a paragraph on that (page 7, line 13),

**This approach of reducing the dynamics of the ocean and the atmosphere to a few spectral large-scale components may at first sight look arbitrary. However for the two midlatitudes basins these modes possess the largest amplitudes (Vannitsem and Ghil, 2017), and for the Tropical Pacific, it is known that these large scale flows are strongly affected by**

the interaction between the ocean and the atmosphere. Moreover we are interested in the basin scale interaction between midlatitudes and the tropics. If such an interaction exists, we expect that these should be visible through the analysis of these large-scale fields. It is clear that these specific variables do not represent the full dynamics, and additional analyses with more modes is worthwile, in particular to see what is the role of the main currents present in the ocean like the Gulf Stream or the Kuroshio current.

*9. The authors use the monthly fields for the analysis. Are these snapshots or monthly averages?*

Yes it is monthly averages. We have clarified that at line 9, page 8, and in the caption of Figure 1.

*10. What was dmin for each time series?*

The value of $d_{min}$ changes for each reference point (on the attractor projection) in the sample of size $L$, since it is the minimum distance among the analogs around the reference point. And of course it also changes with the sample chosen. There is no specific value of $d_{min}$ for a series. To clarify that, we have modified the description at line 10 of page 4 as

**The quantity** $\min d_j$ **denotes the minimum of** $d_j$ **of the** $j = 1, .., E + 1$ **analogs around the reference point** $\vec{X}(t)$**.**

*11. The authors calculate a CI, and there is a growing tendency in the science community to eradicate its use because of its arbitrariness, among other problems. Can the authors instead provide error estimates on their results?*

This is what we have done by randomly selecting new samples of $L$ events a certain number of time and applying the CCM algorithm again. This provides an error estimate of the possible values of $\rho(L)$. We translated that in a CI using the mean and the standard deviation. This is now explained at the end of section 2 (page 6, line 9):

**Note that in order to evaluate the impact of the random sampling of** $L$ **events in the datasets, we can repeat the sampling a certain number of times and infer a mean (or a median when strong assymetries are present) and a standard deviation (here using the Fisher Z test). In the experiments that will be described below, this approach is adopted and each correlation value is estimated over a large number of samples. The algorithm is sketched in Appendix A.**

*12. It is important that the analogs are independent from each other. How did the authors ensure that? Please at least discuss this.*

We tested different period of exclusions when selecting the analogs. We chose 48, 24 and 12 months. Since there were no substantial differences in the results, we decided to keep a minimum of 12 months between analogs. A sentence has been added at page 11, line 8:

**Note also that for this analysis, the analogs have been selected to be at least separated by a period of 12 months. Longer exclusion periods have been used without substantial differences.**

*13. Please make the legends of all figures larger, and the error estimate lines thicker.*

We have increased the size of the legends.

*14. Fig 3 (c) and (e) should be interchanged(?)*

Thank you very much for pointing out this. No but the text should be corrected. We have done that at the beginning of page 13.

*15. What happens for low time lags in fig 4f?*

We have now represented the full curves.

*16. Please describe the distribution of the random anomalies in page 13 line 15.*

These random anomalies were simulated assuming a gaussian distribution around each monthly mean. The variance of the distribution is estimated from the anomalies of the corresponding month for all years in the datasets. the information has been incorporated at page 13, line 35:

**These random anomalies were simulated assuming a gaussian distribution around each monthly mean. The variance of the distribution is estimated using the anomalies of the corresponding month for all years in the datasets.**

*17. page 13 lines 18-24. I don't necessarily agree with the authors' conclusion on the fact that the annual surrogates give the same correlations as the full time series. This does not mean that there is no underlying dynamical causal structure, only that the annual signal is dominant. One has to be careful, as the annual signal is very, very strong. While the authors argue that they don't want to try to remove the annual signal from the time series because of the issues with doing that, perhaps they have to do it this time, e.g. by using different methods and compare results.*

Thank you very much for this interesting point. We have analyzed using the CCM the monthly anomalies discussed above. The results are displayed in Figs. 2 and 3. These results confirm what was said when analyzing the surrogates.

One very interesting result (corroborating what we said when looking at the surrogates) is the fact that there is no influence between the anomalies over the Tropical Pacific and the North Atlantic for both reanalysis datasets. See panels (a) and (c) of each figure.

These results are now incoporated in the text as follows.

In section 4.1, at page 15, line 17:

**Finally to further test the robustness of these results a complementary way to clarify whether montlhy anomalies between different basins are indeed related to each other is to apply direcly CCM on these anomalies. The results are displayed in the Appendix C, Fig. C1. The same conclusions are reached with the absence of dependences between the variables over the North Atlantic basin and the Tropical Pacific,**

[Figure]

Figure 2: CCM as a function of the length L of the samples, as obtained from the anomaly of the monthly time series displayed in Fig. 1 for the reanalyses ERA-20C and ORA-20C. Each line with symbols corresponds to the influence of one variable on a specific coupled ocean-atmosphere basin. The specific variables are denoted in the labeling of each line in each Panel.

[Figure]

Figure 3: CCM as a function of the length L of the samples, as obtained from the anomaly of the monthly time series displayed in Fig. 2 for the reanalyses ERA-20C and ORAS4. Each line with symbols corresponds to the influence of one variable on a specific coupled ocean-atmosphere basin. The specific variables are denoted in the labeling of each line in each Panel.

and a strong mutual dependence between the ocean dynamics over the North Atlantic and the North Pacific.

And in section 4.2, at page 17, line 18:

Finally for the sake of completeness, the application of the CCM to the monthly anomalies is displayed in Fig. C2. Here as for the ERA-20C/ORA-20C dataset, no link between the anomalies of the Tropical Pacific and the North Atlantic is found (panels (a) and (c)). An important difference is however visible on the influence of the North Pacific ocean temperature on the North Atlantic and on the Tropical Pacific (panels (b) and (d)). Again, this contrasts with the results obtained with the other reanalysis dataset.

*18. section 4.2. I'm not sure one can say that ORA-20C is more reliable than ORAS4. I understand from their description that both do not assimilate ocean variables, they only differ in their atmospheric forces. But the ocean circulation is not only determined by the surface forcing, but also, and perhaps to a much larger extend, on the initial condition and on model biases. How do the initial conditions differ between the two ocean products, and assuming they are both initialised from a smoothed observational data set of T and S observations, could it be the case that the ORA-20C run has a less realistic ocean vertical structure because ocean model biases have degraded that longer compared to ORAS4?*

This paragraph has been modified (page 17, line 10) as,

We may conjecture that the ORA-20C reanalysis data set is more reliable since a more consistent atmospheric forcing has been applied and the ocean model has gotten more time to equilibrate around its climate. But care should be taken here is drawing definitive conclusions on that. A better approach to disentangle which of these reanalyses provide the correct answer is to investigate a full coupled ocean-atmosphere reanalysis obtained for the whole 20th Century.

*19. The conclusions are too strong given the CCM method used, the high simplification of the Earth system dynamics to a few in my view quite unrealistic modes, apart from the other potential problems mentioned above. The first conclusion especially, see point 17 above, but also the 4th conclusion. The authors seem to miss the possibility that the atmospheric forcing needs some time to give visible changes in the ocean, e.g. the gyre circulation needs a few years to adapt to a change in the wind stress. It is not necessarily ocean only dynamics that sets time scales of over a year.*

The conclusions have been amended. First a new item is introduced mentioning the different results for the coupling between the North Pacific and the other basins obtained with the two different ocean reanalyses (page 20, line 8),

The results presented here for the two ocean reanalysis datasets disagree on the nature of the dependence between the North Pacific basin and the other ones. The ORA-20C indicates a dynamical influence (transport), while the ORAS4 suggests more an influence dominated by the ocean temperature. This difference is probably related to the specific data assimilation setup used in each case. To clarify what is really at work here new reanalysis datasets are needed. We can conjecture that the most reliable one should be built by using a coupled

**ocean-atmosphere data assimilation system running continuously on the longest period possible.**

and also the last item has been modified as follows,

**The inter-dependences between the North Atlantic and the North Pacific on longer time scales than a year seems to be important, and is probably related to the coupled ocean-atmosphere dynamics on long time scales. One could conjecture that the thermohaline circulation should play a role on this link. Additional analyses with longer data sets, with climate model runs, but also with the analysis of additional basins like the Tropical Atlantic or the Indian Monsoon region are necessary to clarify this role.**

*20. The fact that the CCM method is useful for the simplified model in the appendix does not provide a trong argument that it will work for the real ocean as the simplified model does indeed only have a few modes, while the real Earth system has many, with interactions that are much more complicated and more nonlinear. Please provide a stronger justification on why this experiment backs up the analysis on the far more complicated real system.*

Well if it does not work in such a case then there is no reason to think that it will work on real data. This is why we did it, to get some confidence in the method. Now the application to more sophisticated systems is planned and we add a comment on that at page 6, line 4:

**It provides some confidence in the CCM algorithm, but we should keep in mind that the system explored in Appendix B is relatively simple and the application of CCM on more sophisticated climate models is worth performing. This is left for a future study whose results will be compared with the ones of the present analysis.**

**3    Editor's comments**

We thank the editor for his general comments that are italicised in the following. The modifications in the text are in blue in the attached manuscript.

*1. Motivation and Scope*
*The manuscript Causal dependences between the coupled ocean-atmosphere dynamics over the Tropical Pacific, the North Pacific and the North Atlantic definitely addresses a pertinent problem of interest to Earth System Dynamics, and it does so with a technical treatment grounded on Applied Mathematics. Therefore, the rationale behind the contribution falls within the scope of the journal.*

*However, it should be noted that the manuscript actually tackles a form of dynamic codependence, which is not necessarily equivalent to causal dependence, as shall be explained throughout this document.*

As a general response, we would like to indicate that we do not want to address the complicated problem of tackling the notion of "causality" in all its richness. Rather we adopt a dynamical system approach in which causality has a very precise meaning (see beginning of section 2). This working definition allows to get clear information on the link between variables (or observables), following the works done in the past on that topic, starting from the seminal work of Granger ( e.g.

see his Nobel lecture or his seminal paper of 1969). And we believe that we do not have to modify the terminology in the present analysis of causality since there will be a very important risk to make the reader very much confused on the relation of the present work to the past ones.

Discussing the different notions of causality is certainly very interesting from a philosophical and epistemological point of view but is much beyond the scope of the present investigation. We address below all the specific points raised by the editor.

*2. Methodological Core*

*The literature statement that observables are causally connected if belonging to the same dynamical sys- tem is devoid of physical grounds. A straightforward counter-example stems from fundamental Physics: in Hamiltonian systems there is no causal connection between the intervening variables, notwithstanding their membership to the same dynamical system.*

*The shared membership is solely a proof that the observables have a dynamic codependence, connected through the kinematic-geometry of the dynamical system. Any statements about causality are simply working assumptions guiding the scientific endeavour to formulate informed hypothesis about the system. This being said, the authors have made efforts to avoid the temptation to assign cause-effect relationships between variables belonging to the same dynamical system, i.e. to the same deterministic prescription of their time variation and state-geometric dependence. Rather, the authors address codependence, a legitimate concept when discussing Dynamical Systems, and which entails kinematic-geometric mutual information rather than causality.*

*In fact, the dependence is aptly described as being bilateral and fundamentally cross-inferential, in that knowledge gathered from one observable provides information about others living in the same system. Naturally, this is essentially the case for fully coupled dynamical systems where all variables are actually connected. However, not all dynamical systems are coupled, for which reason the readers should be further cautioned about the detailed assumptions of the methodological construct. A very good service was already done regarding the distinction between unforced and forced systems, so the authors shall easily complement their exposition with the aforementioned aspects.*

The causal dependence which is discussed in the present manuscript is clearly stated at the beginning of section 2. Causal dependence is present provided that the variables share a common attractor manifold and reflects links between variables. This is fully in line with the typical definition one can find in common dictionaries.

We do not see to what the editor is referring to when mentioning the notion of 'causality', and which is probably related to another definition of causality (concept not explained by the editor). In all the papers on the development of CCM, the "co-dependence" (to use the words of the editor) is referred to as "causality" or "causation". See Sugihara et al (2012), Tsonis et al (2015)... If in the present manuscript we change the wording by using "codependence" it is even more confusing than using the concept of causality introduced previously in the literature, starting from the work of Granger (1969). Even if the words could be confusing, we build on past works, and clarifying that wording would need a deep work to understand what is "true causation" and in what context.

*3. Dynamic Dependence vs. Causality*

*This being said, there is a significant risk that the readership will confuse the undeniable inferential power of dynamical system approaches with the concept of causality. For that reason, any statements about observables influencing each other should be avoided unless based on the authors physical interpretation of the specific systems being discussed from their background knowledge.*

*Providing information about (i.e. being a predictor of ) something entails inferential power that may or may not be accompanied by a cause-effect relation. The authors do a good service in staying within the inferential view by saying that codependent variables living in the same dynamical system provide information about each other.*

*Traditional Dynamical System frameworks provide invaluable services to describe system dynamics un- der very specific assumptions stemming from analytical mechanics and kinematic geometry. The readers should be reminded of those assumptions, so that it becomes clear that Dynamical System based diag- nostics are fundamentally motion-descriptive. In this sense, a dynamic coupling between observables is fundamentally a local kinematic-geometric correlation among contemporaries. Therefore, no causality can be inferred from such diagnostic.*

*Overall, a diagnostic of null dynamic dependence does not necessarily preclude the existence of causal dependence. Likewise, the detection of dynamic dependence does not necessarily prove the existence of causality in the system.*

*Focus should be given on whether this particular method has detected or not dynamic codependence among the study data, for the set of conditions under which the method holds, and for the data features that have been actually captured. The causality considerations then come from the authors physical interpretation of the mechanisms at play, as the geometry of the method per se is inherently non-causal.*

Again the concept of causality used here is well stated at the beginning of section 2 (the two first paragraphs) and what is said by the editor is probably related to another concept of causality that we are not addressing here. See also the response to comments 1 and 2.

*4. CCM vs. Alternative Methods*

*The CCM method falls into the usual paradigms of inference from shared information, akin to the rationale behind the classical Mutual Information diagnostics in Information Theory (IT). Mutual In- formation in IT, as CCM, assesses non-directional similarity among observables, yielding positive results when they share information in the multivariate space spun by their dynamics, with the fundamental difference that in IT the treatment is made in the probabilistic space rather than the state space. Therefore, both Mutual Information and CCM fall onto the same paradigm of cross-inference that quantifies the ability to provide information about one observable from another, but contain no proof of causal relation between them.*

*It might then be wondered whether popular measures of directional information (e.g. Conditional Mutual Information, Transfer Entropy, Bayesian Approaches) then provide causal information. Transfer Entropy is by the way the non-Gaussian generalisation of Granger Causality. However, notwithstanding their clear directional inferential power, these are not causal either, as predictors are not necessarily causes and the physics are entirely lost in the statistical constructs undergone in the aforementioned methods. Overall, while comparison with a vast diversity of other methodologies alleging causal inference would be interesting, in reality none of the many published alternatives provides satisfactory treatment of causality either,*

*so this remains fundamentally an open problem. In that sense, despite being clear that this study it investigating dynamic dependence rather than actual causality, it adds to the crucial debate on these problems and should therefore not be lightly dismissed.*

I do agree that the notion of causality is a complicate matter to define as already stated previously. We do not make any claim that we solved that problem, but rather we work with a very practical way to clarify interdependences, referred to as causality or causation, as mentioned at points 1, 2 and 3.

*5. Correlation vs. Dynamic Relation in Estimation Quality Assessment*

*The authors aptly acknowledge the limitations of the classical state space reconstruction, and wisely look for a methodological improvement to implement in their analysis. The adaptations made to CCM are well motivated and enrich the scientific discourse about such procedures. However, it is not devoid of caveats that need to be taken into consideration. The evaluation of the similarity between actual and estimated value of a variable is done in a statistical correlation-based diagnostic that raises serious questions as to whether the quality of the estimation is being properly evaluated in that manner.*

*In fact, contrary to what the reader is led to believe from the manuscript (page 4), a high Pearson correlation between actual and estimated values does not necessarily prove that the estimation is good. Likewise, a low (or even null) correlation is not indicative of fundamental dissimilarity between observa- tion and estimation. These caveats stem from that correlation measure being limited in its adequacy to linearly related, normally distributed data, being assessed for statistically aggregate relationships. In other words, statistical metrics such as Pearson Correlation do not provide the dynamical relation between variables mentioned in the manuscript (page 4). Rather, they provide a first-order linear sta- tistical relation between them, aggregated over the domain where the statistic has been drawn.*

Well I disagree with this view. The correlation is very often used in evaluating forecasts at all lead times in order to check their skill. In fact this correlation is another way to evaluate the mean square error as the following relation holds in general

$$\rho(\hat{Y}, Y) = 1 - \frac{<(\hat{Y} - Y)^2>}{2}$$

assuming that the variables are centered and standardized as it is the case here. So there is one-to-one correspondence between the two. So a high correlation is associated with a low value of the mean square error and vice versa. So it is a clear indication of the quality of an inference or a forecast.

Other scores could be used but the most natural ones are the mean square error or the Pearson correlation.

*6. Prediction vs. Estimation*

*The methodology takes aim at analogs found around one variable X(t) to recover the value of another variable Y (t), contemporary to X(t) (page 4, second sentence). However, in the following sentence the authors mention prediction. Physically speaking, a prediction entails the estimation of a future state. Here, however, the reader is informed about an estimation of a contemporary variable, i.e. without lead time. Therefore, the action to predict should rather be phrased as to estimate.*

OK. I do agree with this point. Historically, Granger (1969) was calling that predictions. We modify this terminology by "inference" or "estimate".

*7. Metric for Analogs*

*According to page 3 (bottom), an Euclidean distance is mentioned for analog selection. In smooth flows living in symplectic manifolds that should work well, as there will be local homeomorphisms between the local attractor charts and an Euclidean tangent bundle. However, that is not necessarily the case for real-world applications where such smoothness no longer holds. Euclidean distances should thus be used only under thoroughly justified appropriate conditions.*

Well in the (continuous) state space which is an Euclidean space, the analogs are in general selected based on a specific norm like the Euclidean distance, whatever the complicated manifold on which the system lives.

*8. State vs. Phase Space*

*A State Space comprises the state variables participating in the system. A Phase Space comprises the state variables and the corresponding conjugated momenta. A system with N state variables has a N - dimensional state space, and a 2N -dimensional phase space. For instance, given a state space $RN$ , the corresponding phase space is $R2N = RN$ $RN$ (which by the way is a symplectic manifold). Interchangeable use of state and phase space is thus prone to ambiguity and error: an unfortunate mis- take that has propagated across significant sectors of the scientific literature. With all due personal respect and consideration for illustrious past references from which we have learned so much, we must nevertheless exert objective criticism when something that they have written is unfortunately incorrect. Published mistakes should in no way be deemed correct when these concepts are rigorously defined in Physics, where each has their unique meaning as explained in the previous paragraph.*

*Therefore, for the sake of clarity, physical rigour, and to avoid any ambiguity and further propagation of misinformation, the term Phase Space needs to be reserved solely to the space spun by the state variables and their conjugate momenta. The state spun by the state variables of the system will have to read State Space. Caution must also be exerted when treating the state space per se, as not all observables involved in the dynamics are actually state variables. For instance, fluxes are not state variables despite being ob- servable quantities. The bottom line is that the state space is not necessarily the space spun by all the observables, but rather solely by the actual state variables. Again here, fundamental Physics informs on which variables of a dynamical system are state variables and which ones are not.*

We disagree with the editor on that point. Phase space is used in both cases as already pointed out to reviewer 2 and links to the specific references. We are really reluctant to change the wording on that but we feel forced to do so by the editor. So we change it without any enthusiasm because there is no "error" behind that choice provided the meaning is clear. Moreover the distinction made by the editor disappears when considering the embedding theorem since it is shown that there is an equivalence between the "state" space representation of the dynamics and its representation based on the successive derivatives of the trajectory (Broer and Takens, 2011).

*9. Geophysical Case Study*

*The geophysical case study discussed in the manuscript is very interesting and ensures the goodness of fit within the scope of Earth System Dynamics. Notwithstanding the existence of prior studies discussing the absence of causal relationships*

*across oceanic basins, the present study has the merit to explore dynamic codependence with a methodology that, despite its caveats, still provides insightful food for thought and discussion. Whether or not dynamic codependence is detectable with the used method depends on both the specific abilities of the method and on the nature of the relations being diagnosed. For that reason, it is wise to refer to what exactly is being diagnosed and what is being eluded i.e. not assessed by the method. In this sense, the authors have already undergone significant revision efforts in their author responses to the referee comments. With further reflection and discussion, the case study can be further strengthened, for which reason I encourage the authors to further contextualise their findings in the light of ocean-atmospheric physics, the methodological construct and practical workflow, the details of which should be further discussed as well (see point 10 below).*

OK. We have made considerably efforts (as already recognized by the editor) in improving the "workflow" of the presentation of the technique (see also Appendix A). We have further clarified the information provided by the technique based on a more detailed analysis of the simple coupled ocean-atmosphere system presented in Appendix B:

At page 5, lines 31-37, we have added

**One important result is the ability of the method to isolate dominant links between the projected attractor (the target of the analysis) and specific variables. The nature of a link can sometimes be directly related to terms present in the dynamical equations but not always due to the multivariate construction of the analogs on the projected attractor. Likewise the absence of relationship inferred from the CCM in the present framework does not imply that there could not be some dependences when other projections of the full state space are used. The conclusions reached are therefore dependent on the specific configuration used and other experimental designs are necessary to corroborate the conclusions. This is planned for a future investigation.**

At page 20, lines 27-29, we have added,

**Finally it should be stressed that the specific design of the CCM approach adopted here based on a low-dimensional projection of the full atmosphere-ocean attractor does not allow to have a one-to-one correspondence of the influence of one variable on another. Moreover other subspaces could be used that can provide different results. More variables should be considered such as projections on additional Fourier modes, or by using projections on a few Empirical Orthogonal Functions. These analyses will allow to evaluate the robustness of the present results.**

At page 24 (appendix B), lines 5-10:

**Note that the angle of approach adopted in Section 2 by considering a low-order projection of the full state space as target does not allow to have a detailed information on the nature of the coupling between the variables since what is inferred is a global influence of a variable on a subset of other variables. The analysis also opens new questions on the role of the different variables in this low-order model and the dependences on the specific subspace selected as the target. This problem is out of the scope of the present work but is worth pursuing in the future. We can now proceed with this approach in the context of the datasets**

**presented in Section 3.**

The algorithm of the CCM is also provided in the new Appendix A.

*10. Reproducibility*

*The reproducibility of the study depends on the readers ability to read in between the lines. However, that should not be the case: the workflow should be fully explicit, otherwise a significant segment of the readership will be alienated. While I was personally able to attest the reproducibility in my own terms, I am skeptical about whether the general readership will. In fact, a thoroughly detailed workflow is still largely missing in the manuscript. A thoroughly detailed workflow with crucial mathematical and algorithmic details will strongly improve the reproducibility and soundness of the manuscript. Only then will be the appealing narrative and results be properly assimilated, and the actual meaning of the diagnostics being undertaken will be properly understood by the readers.*

We have addressed this by revisiting the results of Appendix B and giving some caution at the end of section 2 and in the conclusion as indicated at comment 9 of the editor.

A workflow is also provided in a new Appendix A.

*11. Overall Decision*

*All in all, the manuscript addresses dynamic codependence on the basis of shared attractor membership, rather than causal dependence. For that reason, the title and body should be adjusted accordingly, in order to avoid misinterpretation.*

We will not change the title and the text along these lines since it will introduce additional confusions as discussed at points 1, 2 and 3 above. We hope that the editor will understand our position on that matter.

We have considerably clarified the technique and its limitations as requested by the editor and Reviewer 2.

**References**

[Broer and Takens (2011)] Broer, H., and F. Takens, *Dynamical systems and chaos*, Springer, New York, 2011.

[revised manuscript text omitted]

---

## Editor Decision (ED1)

**ESD-2018-3 Editorial Decision Letter (ESD)**

**Rui A. P. Perdigão**

Vienna, April 17$^{\text{th}}$, 2018

Dear Authors,

Thank you very much for your responses to the referee comments in the interactive discussion, and for having taken my preliminary access review into consideration when uploading the discussion manuscript.

I have high consideration for the diligent efforts undergone by the authors to address a highly relevant and timely problem, and to honestly communicate their work in faithful deference to their background.

Further to the referee reports containing pertinent remarks, and to the authors' responses and diligences, I hereby issue my editorial evaluation with additional aspects for due consideration. I trust that this decision letter will be taken in by the authors as the constructive assessment that it is meant to be.

For ease of readability, this decision letter modularises the evaluation into the following points:

**1. Motivation and Scope**

The manuscript "Causal dependences between the coupled ocean-atmosphere dynamics over the Tropical Pacific, the North Pacific and the North Atlantic" definitely addresses a pertinent problem of interest to Earth System Dynamics, and it does so with a technical treatment grounded on Applied Mathematics. Therefore, the rationale behind the contribution falls within the scope of the journal.

However, it should be noted that the manuscript actually tackles a form of *dynamic codependence*, which is not necessarily equivalent to *causal dependence*, as shall be explained throughout this document.

**2. Methodological Core**

The literature statement that observables are causally connected if belonging to the same dynamical system is devoid of physical grounds. A straightforward counter-example stems from fundamental Physics: in Hamiltonian systems there is no causal connection between the intervening variables, notwithstanding their membership to the same dynamical system.

The shared membership is solely a proof that the observables have a dynamic codependence, connected through the kinematic-geometry of the dynamical system. Any statements about causality are simply *working assumptions* guiding the scientific endeavour to formulate informed hypothesis about the system.

This being said, the authors have made efforts to avoid the temptation to assign cause-effect relationships between variables belonging to the same dynamical system, i.e. to the same deterministic prescription of their time variation and state-geometric dependence. Rather, the authors address *codependence*, a legitimate concept when discussing Dynamical Systems, and which entails kinematic-geometric mutual information rather than causality.

In fact, the dependence is aptly described as being bilateral and fundamentally cross-inferential, in that knowledge gathered from one observable provides information about others living in the same system. Naturally, this is essentially the case for fully coupled dynamical systems where all variables are actually connected. However, not all dynamical systems are coupled, for which reason the readers should be further cautioned about the detailed assumptions of the methodological construct. A very good service was already done regarding the distinction between unforced and forced systems, so the authors shall easily complement their exposition with the aforementioned aspects.

*3. Dynamic Dependence vs. Causality*

This being said, there is a significant risk that the readership will confuse the undeniable inferential power of dynamical system approaches with the concept of causality. For that reason, any statements about observables *influencing* each other should be avoided unless based on the authors' physical interpretation of the specific systems being discussed from their background knowledge.

*Providing information about* (i.e. *being a predictor of*) something entails inferential power that may or may not be accompanied by a cause-effect relation. The authors do a good service in staying within the inferential view by saying that codependent variables living in the same dynamical system *provide information about* each other.

Traditional Dynamical System frameworks provide invaluable services to describe system dynamics under very specific assumptions stemming from analytical mechanics and kinematic geometry. The readers should be reminded of those assumptions, so that it becomes clear that Dynamical System based diagnostics are fundamentally motion-descriptive. In this sense, a dynamic coupling between observables is fundamentally a local kinematic-geometric correlation among contemporaries. Therefore, no causality can be inferred from such diagnostic.

Overall, a diagnostic of null dynamic dependence does not necessarily preclude the existence of causal dependence. Likewise, the detection of dynamic dependence does not necessarily prove the existence of causality in the system.

Focus should be given on whether this particular method has detected or not dynamic codependence among the study data, for the set of conditions under which the method holds, and for the data features that have been actually captured. The causality considerations then come from the authors' physical interpretation of the mechanisms at play, as the geometry of the method per se is inherently non-causal.

*4. CCM vs. Alternative Methods*

The CCM method falls into the usual paradigms of inference from shared information, akin to the rationale behind the classical Mutual Information diagnostics in Information Theory (IT). Mutual Information in IT, as CCM, assesses non-directional similarity among observables, yielding positive results when they share information in the multivariate space spun by their dynamics, with the fundamental difference that in IT the treatment is made in the probabilistic space rather than the state space.

Therefore, both Mutual Information and CCM fall onto the same paradigm of cross-inference that quantifies the ability to provide information about one observable from another, but contain no proof of causal relation between them.

It might then be wondered whether popular measures of directional information (e.g. Conditional Mutual Information, Transfer Entropy, Bayesian Approaches) then provide causal information. Transfer Entropy is by the way the non-Gaussian generalisation of "Granger Causality". However, notwithstanding their clear directional inferential power, these are not causal either, as predictors are not necessarily causes and the physics are entirely lost in the statistical constructs undergone in the aforementioned methods.

Overall, while comparison with a vast diversity of other methodologies alleging causal inference would be interesting, in reality none of the many published alternatives provides satisfactory treatment of causality either, so this remains fundamentally an open problem. In that sense, despite being clear that this study it investigating dynamic dependence rather than actual causality, it adds to the crucial debate on these problems and should therefore not be lightly dismissed.

*5. Correlation vs. Dynamic Relation in Estimation Quality Assessment*

The authors aptly acknowledge the limitations of the classical state space reconstruction, and wisely look for a methodological improvement to implement in their analysis. The adaptations made to CCM are well motivated and enrich the scientific discourse about such procedures.

However, it is not devoid of caveats that need to be taken into consideration. The evaluation of the similarity between actual and estimated value of a variable is done in a statistical correlation-based diagnostic that raises serious questions as to whether the quality of the estimation is being properly evaluated in that manner.

In fact, contrary to what the reader is led to believe from the manuscript (page 4), a high Pearson correlation between actual and estimated values does not necessarily prove that the estimation is good. Likewise, a low (or even null) correlation is not indicative of fundamental dissimilarity between observation and estimation. These caveats stem from that correlation measure being limited in its adequacy to linearly related, normally distributed data, being assessed for statistically aggregate relationships.

In other words, statistical metrics such as Pearson Correlation do not provide the dynamical relation between variables mentioned in the manuscript (page 4). Rather, they provide a first-order linear statistical relation between them, aggregated over the domain where the statistic has been drawn.

*6. Prediction vs. Estimation*

The methodology takes aim at analogs found around one variable $\mathbf{X}(t)$ to recover the value of another variable $Y(t)$, contemporary to $\mathbf{X}(t)$ (page 4, second sentence). However, in the following sentence the authors mention *prediction*. Physically speaking, a prediction entails the estimation of a future state. Here, however, the reader is informed about an *estimation* of a contemporary variable, i.e. without lead time. Therefore, the action to "predict" should rather be phrased as to "estimate".

*7. Metric for Analogs*

According to page 3 (bottom), an Euclidean distance is mentioned for analog selection. In smooth flows living in symplectic manifolds that should work well, as there will be local homeomorphisms between the local attractor charts and an Euclidean tangent bundle. However, that is not necessarily the case for real-world applications where such smoothness no longer holds. Euclidean distances should thus be used only under thoroughly justified appropriate conditions.

*8. State vs. Phase Space*

A *State Space* comprises the *state variables* participating in the system. A *Phase Space* comprises the state variables and the corresponding *conjugated momenta*. A system with $N$ state variables has a $N$-dimensional state space, and a $2N$-dimensional phase space. For instance, given a state space $\mathbb{R}^N$, the corresponding phase space is $\mathbb{R}^{2N} = \mathbb{R}^N \times \mathbb{R}^N$ (which by the way is a symplectic manifold).

Interchangeable use of state and phase space is thus prone to ambiguity and error: an unfortunate mistake that has propagated across significant sectors of the scientific literature. With all due personal respect and consideration for illustrious past references from which we have learned so much, we must nevertheless exert objective criticism when something that they have written is unfortunately incorrect. Published mistakes should in no way be deemed correct when these concepts are rigorously defined in Physics, where each has their unique meaning as explained in the previous paragraph.

Therefore, for the sake of clarity, physical rigour, and to avoid any ambiguity and further propagation of misinformation, the term *Phase Space* needs to be reserved solely to the space spun by the state variables and their conjugate momenta. The state spun by the state variables of the system will have to read *State Space*.

Caution must also be exerted when treating the state space per se, as not all observables involved in the dynamics are actually state variables. For instance, fluxes are not state variables despite being observable quantities. The bottom line is that the state space is not necessarily the space spun by all the observables, but rather solely by the actual state variables. Again here, fundamental Physics informs on which variables of a dynamical system are state variables and which ones are not.

**9. Geophysical Case Study**

The geophysical case study discussed in the manuscript is very interesting and ensures the goodness of fit within the scope of Earth System Dynamics. Notwithstanding the existence of prior studies discussing the absence of causal relationships across oceanic basins, the present study has the merit to explore dynamic codependence with a methodology that, despite its caveats, still provides insightful food for thought and discussion.

Whether or not dynamic codependence is detectable with the used method depends on both the specific abilities of the method and on the nature of the relations being diagnosed. For that reason, it is wise to refer to what exactly is being diagnosed and what is being eluded i.e. not assessed by the method. In this sense, the authors have already undergone significant revision efforts in their author responses to the referee comments.

With further reflection and discussion, the case study can be further strengthened, for which reason I encourage the authors to further contextualise their findings in the light of ocean-atmospheric physics, the methodological construct and practical workflow, the details of which should be further discussed as well (see point 10 below).

**10. Reproducibility**

The reproducibility of the study depends on the reader's ability to read in between the lines. However, that should not be the case: the workflow should be fully explicit, otherwise a significant segment of the readership will be alienated.

While I was personally able to attest the reproducibility in my own terms, I am skeptical about whether the general readership will. In fact, a thoroughly detailed workflow is still largely missing in the manuscript.

A thoroughly detailed workflow with crucial mathematical and algorithmic details will strongly improve the reproducibility and soundness of the manuscript. Only then will be the appealing narrative and results be properly assimilated, and the actual meaning of the diagnostics being undertaken will be properly understood by the readers.

**11. Overall Decision**

All in all, the manuscript addresses dynamic codependence on the basis of shared attractor membership, rather than causal dependence. For that reason, the title and body should be adjusted accordingly, in order to avoid misinterpretation.

The manuscript should thus be revised taking into consideration the referee concerns complemented by the aspects raised in the aforementioned points in the present editorial report. The revision diligences already undergone by the authors in response to the referee reports are encouraging and should be furthered in accordance with the raised concerns.

I have confidence in the authors' ability and willingness to take the referee and editorial remarks into due consideration when proceeding with their manuscript revision.

For that reason, I return the manuscript to the authors, looking forward to the next revised version.

With very best wishes,

Rui Perdigão
(ESD Editor)

———

Reference of this letter:

Perdigão, R.A.P. (2018): ESD-2018-3 Editorial Decision Letter, 17.04.2018, Earth System Dynamics. To be available at URL: https://www.earth-syst-dynam-discuss.net/esd-2018-3/ upon eventual final publication of ESD-2018-3 at Earth System Dynamics (Vannitsem and Ekelmans, in Review).

Reference of the ESD-2018-3 manuscript at Earth System Dynamics - Discussions:

Vannitsem, S. and Ekelmans, P.: Causal dependences between the coupled ocean-atmosphere dynamics over the Tropical Pacific, the North Pacific and the North Atlantic, Earth Syst. Dynam. Discuss., https://doi.org/10.5194/esd-2018-3, in review, 2018.

---

## Author Response (AR2)

**Response to the comments of the editor**

July 26, 2018

**Stéphane Vannitsem[1] and Pierre Ekelmans[2]**

[1]Royal Meteorological Institute of Belgium, Avenue Circulaire, 3, 1180 Brussels, Belgium

[2]Theory of Neural Dynamics Group, Max Planck Institute for Brain Research, Max-von-Laue-Strasse 4, 60438 Frankfurt, Germany

Dear Editor

We would like first to thank you for the very constructive comments that you have made during this second round of review. Please find below how we address your three different points that have to be settled. The additional text is colored in green in the pdf version of the new manuscript.

*1. Causality: There is a clash of cultures between physical and the kinematic-geometric views. In fundamental Physics and most fields of natural science, causality pertains cause-effect relationships. This is what had been meant as causality in the previous editorial report. In kinematic-geometric approaches to dynamical systems, there is no proven cause-effect among co-members of a dynamical system, but rather a codependence bond as expressed through kinematic-geometric couplings, which may be physically non-causal. The term causation mentioned in the author response is yet another concept that is physically different from that of causality but is not the object of study in the manuscript, therefore the only note I leave in that regard is that these are not equivalent concepts. Closing the matters on point 1 , the conceptual debate on causality can be avoided in the revised manuscript by clearly reiterating that the work addresses causality from a dynamical systems perspec- tive, and acknowledging that this is not to be confused with the physical cause-effect definition. This way, causality statements on the manuscript will be positioned on their specific kinematic-geometric context, deploying the analysis and discussion without conceptual clashes.*

We indeed do agree with your point. The notion of causality is multifaceted and it is not easy to grasp this concept in one definition. In order to clarify the specific definition used here, we have introduced the following text at page 3, line 23:

**It must be stressed here that the notion of causality is used in the very specific context of dynamical systems theory discussed above, that should not be confused with the traditional cause-effect relationship. From now on, the words causality and dependences will be used equivalently.**

We have also stressed the specific notion used here by introducing at several places in the introduction the terms **in a dynamical sense** when speaking of causality.

*3. Correlative measures used to validate results, while still widely popular in some fields (e.g. in weather forecasting), are not devoid of caveats as noted in previous referee and editorial reports. However, such measures can be used provided that their limitations are clear and their power is not overstated. All measures have caveats and it is always beneficial to have them at least briefly stated in the text so that the readers will not get overconfident on the actual value of the diagnostics being made. This will settle point 3.*

We also agree on that. We have introduced a comment as a footnote in the description of the technique in order to avoid breaking the readability of the description. This is introduced at page 4 as

**Note again that the correlation is measuring the linear association that could sometimes be insufficient in the case of nonlinear dependences**

*4. Euclidean metrics should not be assumed as straightforwardly valid, instead being justified in the light of the problem being studied. Their use can be justified when the smoothness of the dynamics is ensured to enable a homeomorphic mapping between their kinematic geometry and that of the tangent space where the Euclidean metrics are usually deployed. While not all dynamic processes will produce a valid phase space manifold, in many geophysical fluid dynamics applications that working assumption can be justified to some extent in light of the nature of the flow, provided that special care is taken when dealing with discretisation issues naturally emerging from operational finite-step treatments (be they data-based or numerical). By explicitly stating the qualifying attributes (e.g. smoothness of the dynamical system under study), the authors can then provide a geophysically-satisfactory justification for the use of the chosen metric. This will settle point 4.*

We introduced a comment on that aspect at page 4, line 3 as

**This can be done provided the solutions of the dynamical system display sufficiently smooth properties (e.g. continuity) as it is often assumed for large scale geophysical fluid dynamics.**

---

## Editor Decision (ED2)

**ESD-2018-3 Editorial Decision Letter (ESD)**

**Rui A. P. Perdigão**

Vienna, July 9th, 2018

Dear Authors,

Thank you for your diligent efforts addressing the raised concerns and producing an improved version. The revised manuscript has been sent for peer-review, resulting in a reiteration of some of the concerns raised previously. While I cannot endorse the tenor of communication, I would still kindly draw the authors' attention to the scientific content of the report and overall peer-review process for consideration.

Taking all elements into consideration with scientific plurality without loss of rigour, the following comments shall further enable a swift and effective closure of the pre-publication review process and debate:

**1) Causality:** There is a "clash of cultures" between physical and the kinematic-geometric views. In fundamental Physics and most fields of natural science, causality pertains cause-effect relationships. This is what had been meant as causality in the previous editorial report. In kinematic-geometric approaches to dynamical systems, there is no proven cause-effect among co-members of a dynamical system, but rather a codependence bond as expressed through kinematic-geometric couplings, which may be physically non-causal.

The term causation mentioned in the author response is yet another concept that is physically different from that of causality but is not the object of study in the manuscript, therefore the only note I leave in that regard is that these are not equivalent concepts.

***Closing the matters on point 1***, the conceptual debate on causality can be avoided in the revised manuscript by clearly reiterating that the work addresses causality *"from a dynamical systems perspective"*, and acknowledging that this is not to be confused with the physical cause-effect definition. This way, causality statements on the manuscript will be positioned on their specific kinematic-geometric context, deploying the analysis and discussion without conceptual clashes.

**2) Granger causality** is an inferential statistical measure popular in applied stochastics. However, it is devoid of fundamental causal diagnosability in Physics and fundamentally differs from the dynamical systems diagnostics. Moreover, as inferential measure it is only valid only in statistical problems with normally distributed random distributions. Therefore, it is neither relevant to physical causality nor to dynamical systems approaches in complex systems and its discussion is thus not worth further pursuing in this study. ***This matter is therefore settled.***

**3) Correlative measures** used to validate results, while still widely popular in some fields (e.g. in weather forecasting), are not devoid of caveats as noted in previous referee and editorial reports. However, such measures can be used provided that their limitations are clear and their power is not overstated. All measures have caveats and it is always beneficial to have them at least briefly stated in the text so that the readers will not get overconfident on the actual value of the diagnostics being made. ***This will settle point 3.***

**4) Euclidean metrics** should not be assumed as straightforwardly valid, instead being justified in the light of the problem being studied. Their use can be justified when the smoothness of the dynamics is ensured to enable a homeomorphic mapping between their kinematic geometry and that of the tangent space where the Euclidean metrics are usually deployed. While not all dynamic processes will produce

a valid phase space manifold, in many geophysical fluid dynamics applications that working assumption can be justified to some extent in light of the nature of the flow, provided that special care is taken when dealing with discretisation issues naturally emerging from operational finite-step treatments (be they data-based or numerical). By explicitly stating the qualifying attributes (e.g. smoothness of the dynamical system under study), the authors can then provide a geophysically-satisfactory justification for the use of the chosen metric. ***This will settle point 4.***

**5) Phase and state spaces:** The authors have changed the notation and that is appreciated. However, given the tone of the author response in this regard and further reiteration from the referee, I believe that an additional clarification is hereby due in order to become clear that this was a constructive scientific remark rather than an editorial choice. As already noted in previous reports, an N-dimensional state space corresponds to a 2N-dimensional phase space (the direct tensor product between the N-dimensional state space and the N-dimensional space of the tendencies). Therefore, there is a conceptual and dimensional inconsistency in interchanging the two notions. ***This matter is therefore settled.***

In the editorial context, my focus is to ensure that the raised scientific and technical concerns are debated and taken into due consideration so that a solid contribution to science ultimately emerges at ESD.

Irrespective of whether one fundamentally agrees or not with the approaches being explored in the paper, and notwithstanding the open debate that could still be pursued relative to the study, the important aspect at this stage is to ensure that that the present investigation and respective arguments are presented in a sound manner that is scientifically consistent, rigorous and reproducible in the field of study, so that the community can further debate and proceed the scientific quest in a thoroughly informed manner.

I thus return the manuscript to the authors for a closing revision, looking forward to a final version.

With best regards,

Rui Perdigão
(ESD Editor)

———

Reference of this letter:

Perdigão, R.A.P.: ESD-2018-3 Editorial Decision Letter, 09.07.2018, Earth System Dynamics. To be available at URL: https://www.earth-syst-dynam-discuss.net/esd-2018-3/ upon eventual final publication of ESD-2018-3 at Earth System Dynamics (Vannitsem and Ekelmans, in Review).

Reference of the ESD-2018-3 manuscript at Earth System Dynamics - Discussions:

Vannitsem, S. and Ekelmans, P.: Causal dependences between the coupled ocean-atmosphere dynamics over the Tropical Pacific, the North Pacific and the North Atlantic, Earth Syst. Dynam. Discuss., https://doi.org/10.5194/esd-2018-3, in review, 2018.